# The mutational impact of culturing human pluripotent and adult stem cells

Ewart Kuijk [1✉], Myrthe Jager [1], Bastiaan van der Roest [1], Mauro D. Locati[1], Arne Van Hoeck [1], Jerome Korzelius [2], Roel Janssen [1], Nicolle Besselink [1], Sander Boymans [1], Ruben van Boxtel [3] & Edwin Cuppen [1✉]

Genetic changes acquired during in vitro culture pose a risk for the successful application of stem cells in regenerative medicine. To assess the genetic risks induced by culturing, we determined all mutations in individual human stem cells by whole genome sequencing. Individual pluripotent, intestinal, and liver stem cells accumulate 3.5 ± 0.5, 7.2 ± 1.1 and 8.3 ± 3.6 base substitutions per population doubling, respectively. The annual in vitro mutation accumulation rate of adult stem cells is nearly 40-fold higher than the in vivo mutation accumulation rate. Mutational signature analysis reveals that in vitro induced mutations are caused by oxidative stress. Reducing oxygen tension in culture lowers the mutational load. We use the mutation rates, spectra, and genomic distribution to model the accumulation of oncogenic mutations during typical in vitro expansion, manipulation or screening experiments using human stem cells. Our study provides empirically defined parameters to assess the mutational risk of stem cell based therapies.

[1] Center for Molecular Medicine and Oncode Institute, University Medical Center Utrecht, Universiteitsweg 100, 3584 CG Utrecht, The Netherlands. [2] Leibniz Institute on Aging—Fritz Lipmann Institute, Beutenbergstraße 11, Jena 07745, Germany. [3] Princess Máxima Center for Pediatric Oncology, Heidelberglaan 25, 3584 CS Utrecht, Utrecht, The Netherlands. ✉email: E.W.Kuijk-3@umcutrecht.nl; ecuppen@umcutrecht.nl

As infinite supplies of undifferentiated and specialized cells, induced pluripotent stem cells (PSCs) and adult stem cell (ASC)-derived organoids hold great potential in regenerative medicine[1,2]. Furthermore, stem cells have become invaluable tools in pharmacology and toxicology for in vitro testing[3–6]. Maintenance of genomic integrity in culture is a prerequisite for the successful application of stem cells, as unwanted mutations may influence drug responses and toxicology measurements, or might lead to oncogenic transformation following transplantation. However, multiple studies have identified recurrent genomic alterations that originate in routinely cultured human ASCs and PSCs[7–20]. Although these studies suggest that the genomic integrity of stem cells is reduced in culture, the majority of these studies have been performed on bulk cultures with methods that have limited resolution. Analysis of bulk samples poses important limitations as only mutations that are shared among the majority of the cells are detectable. This results in a bias toward those genomic alterations that confer a selective advantage upon the cells, or those that occurred early in the culturing process, while the mutational impact of the culturing process itself and the responsible mutational mechanisms remain as yet unclear. A better understanding of the mutational processes that are active in culture might enable us to improve genomic stability in culture by changing the culture conditions.

We recently established a method to accurately identify in vivo-acquired somatic mutations in individual stem cells at base-pair resolution by combining whole-genome sequencing (WGS) with in vitro clonal expansion. In short, stem cells are seeded as single cells and propagated to obtain sufficient cells for DNA isolation and subsequent WGS. Bioinformatic analyses are performed that identify with high confidence those mutations (single-nucleotide variants, indels, copy number alterations, structural variants (SVs), and aneuploidies) present in the original cell and filter out germline variants and subclonal mutations that occurred after the single-cell step[21]. With this method, high noise rates of in vitro amplification approaches are avoided, which enables the reliable detection of less than a hundred mutations per genome[22]. The resultant genome-wide mutation spectra and distribution have been shown to provide novel insights into the activity of specific mutational and DNA repair processes in adult stem cells in vivo and CRISPR/Cas9-edited organoids[23–25].

Here, we have adapted this approach to systematically measure the mutational impact of in vitro culture on individual cells of three human stem cell types that were chosen for their relevance in pharmacology, toxicology, and regenerative medicine: PSCs, liver ASCs, and intestinal ASCs. Our study demonstrates that in vitro culture results in increased mutation rates, and leaves a distinct but common mutational footprint related to oxidative stress that is independent of stem cell type. Furthermore, we demonstrate that culturing under reduced oxygen tension decreases the amount of mutations related to oxidative stress. We used the measured quantitative and qualitative mutational characteristics to model genome-wide mutation accumulation and perform genetic risk assessments associated with in vitro and in vivo applications of stem cells.

## Results

**Mutation accumulation in PSCs and ASCs during culture**. To investigate the mutational consequences of standard culture conditions on the genome of stem cells in an unbiased manner, we first established clonal human PSC and ASC lines. Clones were established for one pluripotent embryonic stem cell line, two induced PSC lines, two liver ASC lines, and two intestinal ASC lines. Each clonal line was cultured for ~2–5 months (Supplementary Data 1), during which mutations were allowed to accumulate. Subsequently, a second clonal step was performed to generate so-called subclones that were used to determine the mutations present in the individual cells that gave rise to these subclones. Clones, subclones, and matched nonclonal reference samples were subjected to WGS (Fig. 1a). All germline variants and variants that accumulated before the first clonal step were excluded from the subclone. Variants that arose in vitro after the second clonal step were filtered out bioinformatically based on low variant allele frequency (VAF). This approach allowed us to specifically measure the mutations that accumulated in the culture period in between the two clonal steps.

The total number of stem cells that is required for a particular application, such as a cell therapy or drug study, is achieved after a certain amount of population doublings, which is defined by the cell division rate and the cell death rate. Thus, from a practical perspective, knowledge on the mutation rates per population doubling is more informative than the number of mutations per time unit. Therefore, we determined the population-doubling rate for all cell types under the same conditions that were also used for the mutation accumulation experiments. We found a population-doubling time of ~23 h for human PSCs, which is approximately twice as fast as for the human liver and intestinal ASCs that have population-doubling times of ~46 and ~44 h, respectively (Fig. 1b).

The karyotypes of all clones and subclones for all three stem cell types were analyzed by sequencing read-depth coverage, and found to be normal without any gross chromosomal abnormalities. In total, we detected 283 small insertions and deletions (indels) in the 15 subclones (Supplementary Data 1). There was no statistically significantly difference between the different stem cell types in the number of indels per genome per population doubling (ANOVA, $p = 0.15$, Fig. 1c). Genomic distribution analysis revealed no effects of the indels on CDS, with the exception of one indel in the subclone of the human embryonic stem cell line that caused a frameshift mutation in *CDKN1C*.

We identified a total of 4,517 single-base-pair substitutions (SBS) unique to the 15 subclones. Liver ASCs acquired $8.3 \pm 3.6$ SBS per genome per population doubling. In intestinal ASCs, the mutation accumulation rate was similar with $7.2 \pm 1.1$ SBS per genome per population doubling. The number of SBS was lower in the PSCs than in the ASCs, with $3.5 \pm 0.5$ mutations per genome per population doubling (ANOVA, $p = 0.006$, Fig. 1d). For comparison with the in vivo mutation rates, we also calculated the rate per year for both ASC types. Individual intestinal ASCs accumulate $1415 \pm 212$ mutations per year, and liver stem cells accumulate $1588 \pm 679$ mutations per year. These values are nearly 40-fold higher than the in vivo mutation accumulation rate of approximately 40 SBS per year[24]. These findings demonstrate that the standard in vitro culture conditions have a high impact on mutation accumulation rates.

SBS was significantly depleted in genic regions for all three stem cell types (Fig. 1e). The depletion was not restricted to the protein-coding sequence, but also included noncoding sequences, indicating that the depletion in genic regions is mainly caused by enhanced repair activity in genic regions and not by selection against deleterious mutations. The depletion in coding sequences was around 2% in all three stem cell types (Fig. 1f). In total, we identified 19 nonsynonymous mutations across all samples, 8 in the PSCs, 2 in the liver ASCs, and 9 in the intestinal ASCs (Supplementary Data 2). None of the nonsynonymous mutations affected known cancer genes based on the census of human cancer genes[26], or have previously been described to confer a selective advantage over stem cells in culture[11,19].

Mutations in promoter regions can affect gene activity and thereby contribute to disease[27,28]. Mutations were depleted in the promoter or promoter-flanking regions[29] of all three stem cell

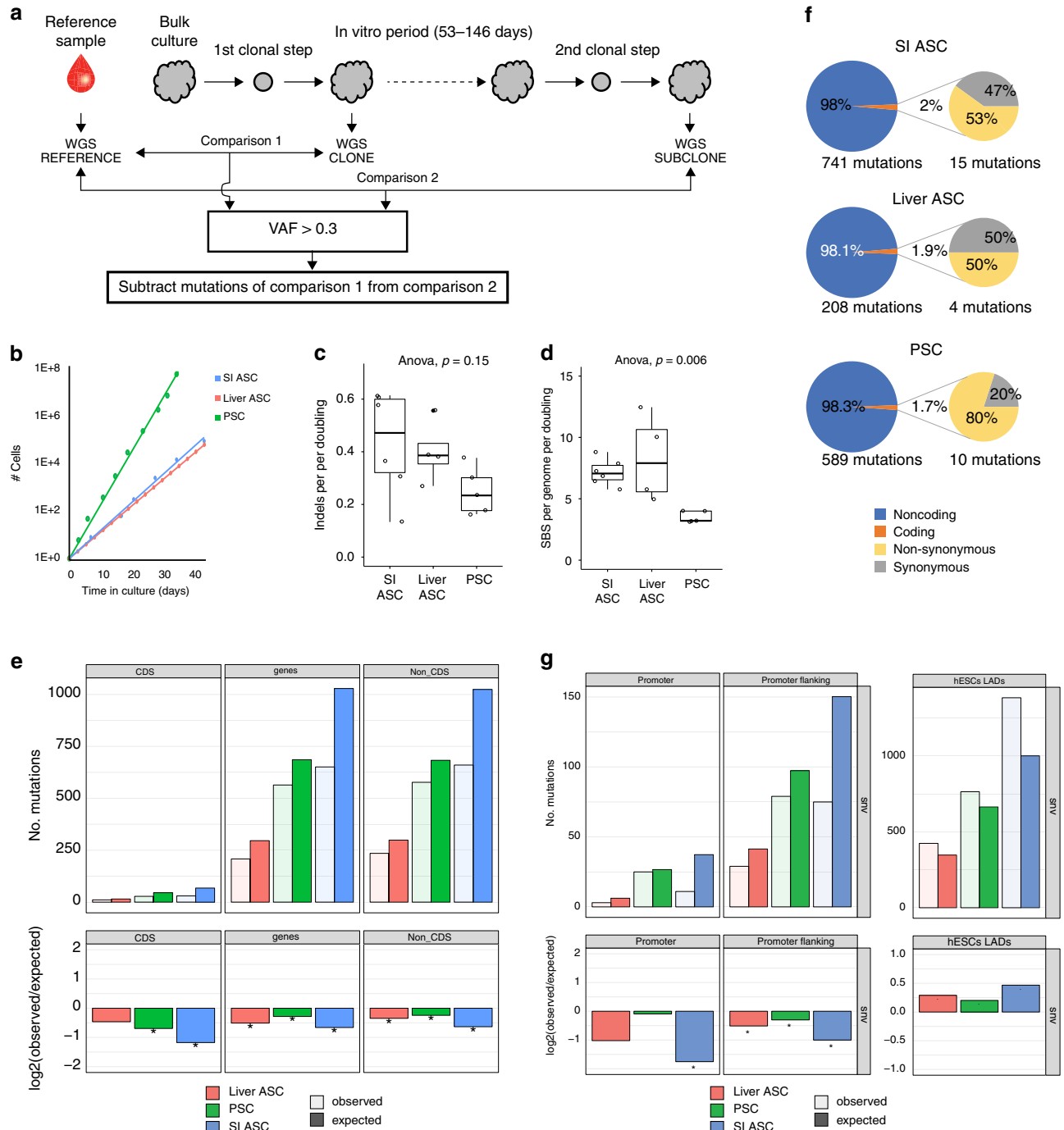

**Fig. 1 In vitro mutation accumulation in PSCs ($n = 5$), liver stem cells ($n = 4$), and intestinal stem cells ($n = 6$). a** Schematic overview of the experimental setup to determine in vitro-accumulated mutations in individual human PSCs, liver stem cells, and intestinal stem cells. Clonal stem cell lines were cultured for ~2–5 months during which mutations were allowed to accumulate. At the end of the culture period, a second clonal step was performed, and the derivative subclones were expanded until enough DNA could be isolated for WGS analysis. A biopsy or bulk culture was used as a reference sample to determine and exclude all germline variants, which were subtracted from the clone (comparison 1) and from the subclone (comparison 2). Finally, with these results, the clonal variants were subtracted to identify all variants that were unique to the subclone. **b** Growth curves for PSCs (green), liver stem cells (red), and intestinal stem cells (blue). **c** Box–whisker plots showing the median, first and third quartiles, and minimum and maximum of the number of indels per genome per population doubling (statistical test used: ANOVA). **d** Box–whisker plots showing the median, first and third quartiles, and minimum and maximum of the number of SBS per genome per population doubling. Statistical test used: ANOVA. **e** Top panels: genomic distribution of the observed versus the expected amount of mutations per genome per doubling. Bottom panels: log2 of the ratio between the total number of observed mutations versus the total number of the expected amount of mutations. Asterisks denote statistically significant differences between observed and expected (binomial test). CDS coding sequence. **f** Relative number of SBS per cell type (left circles) and those affecting protein-coding DNA (right circles). Nonsynonymous SBS leads to amino-acid changes, whereas synonymous mutations do not have an effect on the protein sequence. **g** Top panels: genomic distribution of the observed versus the expected amount of mutations per genome per doubling at promoter and promoter-flanking regions. Error bars represent the standard deviation. Bottom panels: log2 of the ratio between the total number of observed versus the total number of the expected amount of mutations at promoter and promoter-flanking regions. Asterisks denote statistically significant differences between observed and expected.

types, particularly in the intestinal ASCs (Fig. 1g). Mutations in heterochromatin are generally considered harmless, because the risk that functional elements are affected is low. In all three stem cell types, mutations were significantly enriched in heterochromatic laminin-associated domains (LADs) (Fig. 1g), as is the case for mutations that arise during reprogramming[30]. Functionally relevant genomic regions thus appear more protected from mutation accumulation in all three stem cell types, while heterochromatic LADs are more susceptible to mutation accumulation.

**In vitro mutation accumulation is caused by oxidative stress.** To obtain insight into the mutational processes induced by culturing, we analyzed the mutation spectra of the three different stem cell types (Fig. 2a). The in vitro mutation accumulation spectra of liver and intestinal ASCs were significantly different from their in vivo counterparts (Pearson's chi-squared test; liver ASCs: $p$ value $= 2.0e-10$, intestinal ASCs: $p$ value $< 2.2e-16$). The dominant mutation type in intestinal stem cells in vivo is C > T

changes in a CpG context[24], while the contribution of this mutation type to the mutation spectrum of in vitro-cultured intestinal ASCs was low (Fig. 2a). C > T transversions were the predominant base substitutions in the mutational spectrum of all three stem cell types, encompassing nearly 30% of the base substitutions in the liver ASCs, over 35% of all the base substitutions in the intestinal ASCs, and even more than 40% of the SBS in the PSCs (Fig. 2a). This mutation type has been linked to reactive oxygen species (ROS)[31,32]. Previous studies[8,33] have demonstrated that human PSCs are susceptible to oxidative stress-related DNA damage when cultured under atmospheric levels of oxygen (20% $O_2$). To further investigate the effect of oxygen levels on mutation accumulation, we used our experimental setup (Fig. 1a) to measure mutation accumulation in individual cells for three different clonal PSC lines that were cultured for 3 months under reduced oxygen tension (3% $O_2$). In total, 532 SBS were identified that were unique to the subclones. PSCs cultured under reduced oxygen acquired 2.1 ± 0.3 SBS per genome per doubling,

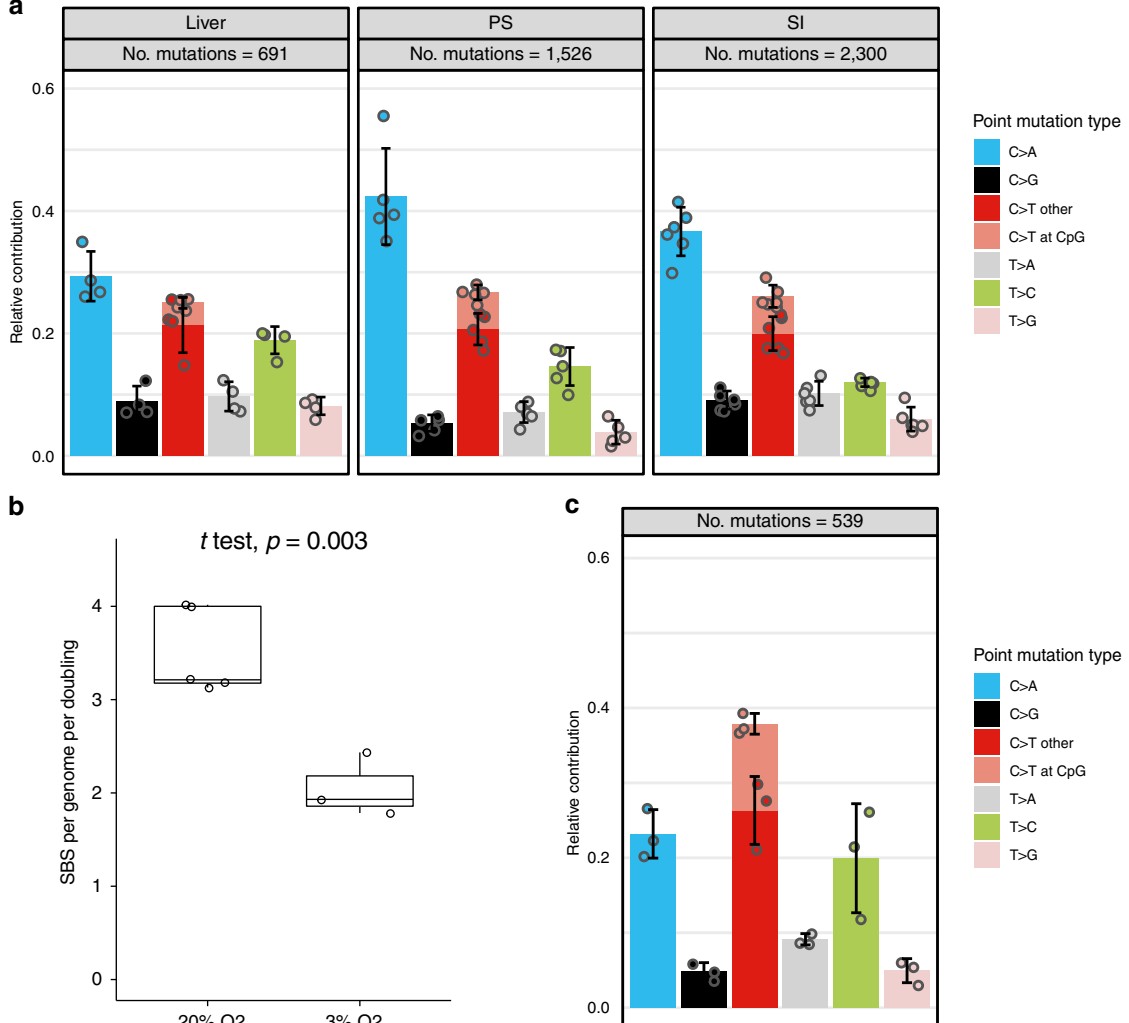

**Fig. 2 Mutational spectrum and signature analysis. Data are derived from biological replicates. a** Relative contribution of the indicated base substitution types to the mutation spectrum. Per stem cell type, data are represented as the mean relative contribution of each mutation type over all subclones (liver $n = 4$, PSC $n = 5$, and SI ASC $n = 6$). Error bars represent the standard deviation from the mean. The total number of SBS is indicated. **b** Box–whisker plots showing the median, first and third quartiles, and minimum and maximum of the number of mutations per genome per population doubling under 20% and 3% $O_2$. Statistical test used: two-sided $t$ test. **c** Relative contribution of the indicated base substitution types to the mutation spectrum of individual human pluripotent stem cell lines ($n = 3$) cultured for 3 months under 3% oxygen levels. Data are represented as the mean relative contribution of each mutation type over all subclones. Error bars represent the standard deviation from the mean. The total number of SBS is indicated.

which is a significant reduction when compared with the PSCs that were cultured under atmospheric oxygen levels (Fig. 2b). The mutational spectrum was also significantly different from the spectrum of PSCs cultured under atmospheric oxygen levels (Pearson's chi-squared test, $p$ value < 2.2e−16). This difference was mainly caused by a significant reduction in the relative number of C > T changes from around 40% to nearly 20%. This coincided with a relative increase in the number of C−T changes, particularly at CpG sites (Fig. 2c). Thus, culturing under reduced oxygen tension lowers the amount of in vitro-induced mutations that are related to oxidative stress.

**Mutational signatures.** To obtain more insight into the mutational processes in cultured cells, we performed a de novo extraction of mutational signatures with nonnegative matrix factorization[34], using the mutational landscapes of all samples, including the PSC cultures under high and low-oxygen levels. We identified three distinct signatures that were designated signatures A–C (Fig. 3a). Signatures A and C were characterized by C > T mutations, while signature B showed relatively more C−G and T−C mutations and less C > T mutations. Unsupervised hierarchical clustering of the samples based on the relative contribution of the extracted signatures to the mutational profile

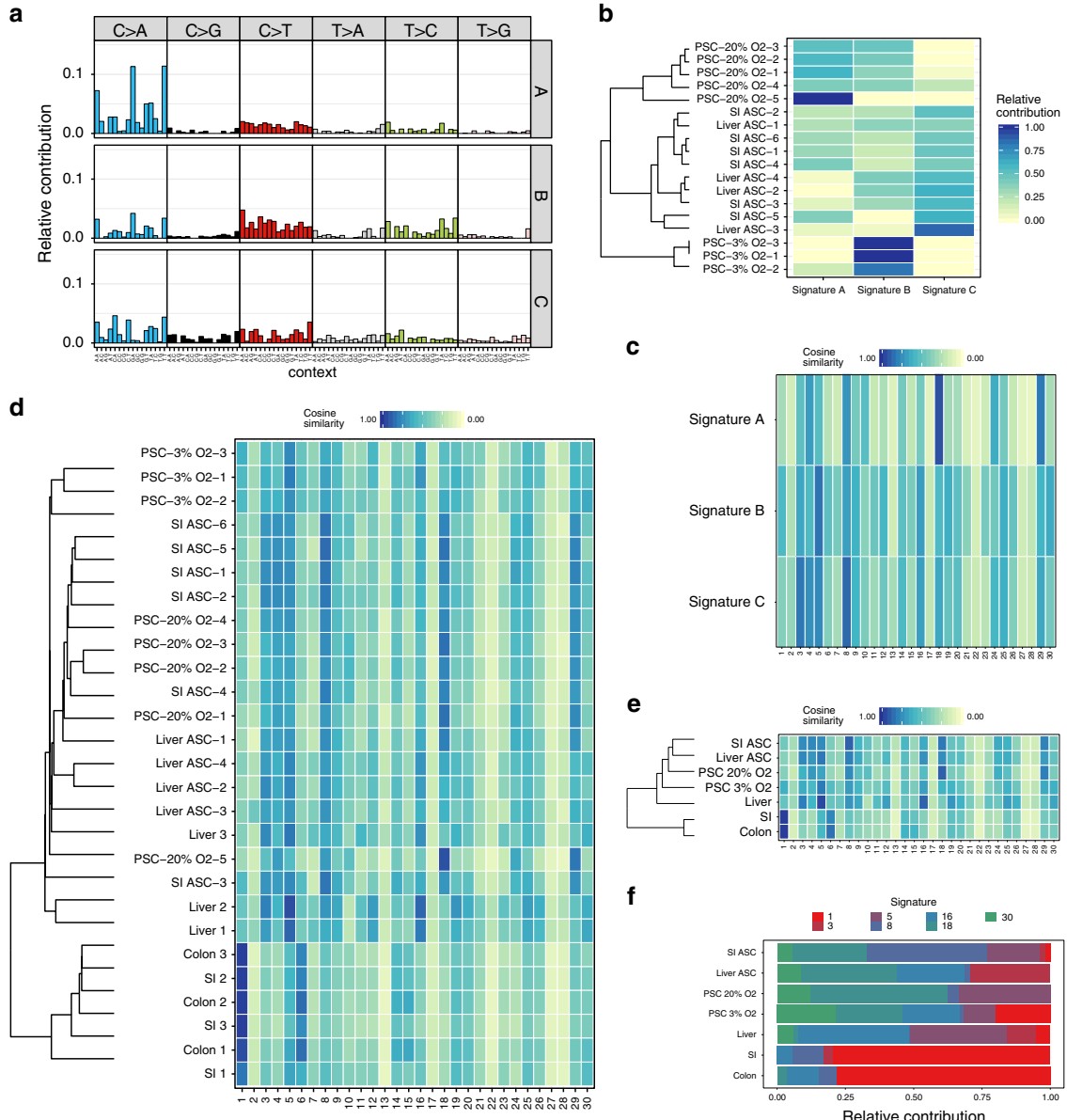

**Fig. 3 Mutational signature analysis. a** The 96-type mutational profiles of signatures A–C extracted by nonnegative matrix factorization from the mutational landscape of all in vitro samples. **b** Cosine similarity heatmap between the mutational profiles of the extracted signatures A–C and the samples. Unsupervised hierarchical clustering of the samples based on the relative contribution of the extracted signatures to the mutational profile resulted in a clear segregation of the samples by cell type. **c** Cosine similarity heatmap between the mutational profiles of the extracted signatures A–C, and the COSMIC Signatures. **e** Cosine similarity heatmap between the COSMIC Signatures and the mutational profiles of the in vitro stem cell types and the previously described in vivo ASCs[24]. Liver = liver stem cells in vivo, SI = intestinal stem cells in vivo, colon = colon stem cells in vivo. Unsupervised hierarchical clustering of the samples was performed based on the cosine of the mutational profile to the COSMIC Signatures. **d** Cosine similarity heatmap between COSMIC Signatures and the mutational profiles of all in vitro stem cell subclones and the previously described in vivo ASCs[24]. Liver = liver stem cells in vivo, SI = intestinal stem cells in vivo, colon = colon stem cells in vivo. Unsupervised hierarchical clustering of the samples was performed based on the cosine of the mutational profile to the COSMIC Signatures. **f** Relative contribution of the COSMIC Signatures to the different stem cell types.

resulted in a clear segregation of the samples by cell type (Fig. 3b). Signature A was enriched in the 20% $O_2$ PSCs, signature B was enriched in the 3% $O_2$ PSCs and to a lesser extent in the 20% $O_2$ PSCs, and signature C was enriched in the liver and intestinal ASCs (Fig. 3a, b). These findings indicate that distinct mutational processes are active in cultured ASCs, 20% $O_2$ PSCs, and 3% $O_2$ PSCs.

We compared the de novo signatures extracted from the in vitro-cultured stem cells with the COSMIC signatures[23]. Signature A highly resembled COSMIC Signature 18, signature B was mostly similar to COSMIC Signature 5, and signature C mostly resembled COSMIC Signature 8 (Fig. 3c). Next, we examined the relative contribution of the COSMIC Signatures to the different stem cell types. As a reference, we also included the previously described mutational profiles of in vivo-accumulated mutations for the liver, the small intestine, and the colon in this analysis[24]. Hierarchical clustering based on similarity with the COSMIC signatures resulted in clustering of the in vitro-cultured samples and within that cluster grouping by stem cell type. The PSCs that were cultured under reduced oxygen levels, were grouped together in a separate cluster (Fig. 3d). A similar pattern was observed when the data were aggregated per cell type, with the in vitro samples cultured under atmospheric oxygen levels clustering together resembling COSMIC Signatures 8 and 18 (Fig. 3e). There was also a relatively high contribution of COSMIC Signatures 8 and 18 to the in vitro-cultured samples, while the contribution of these signatures was less in the cells that were cultured under 3% oxygen, which showed increased contribution of COSMIC Signatures 1 and 30 (Fig. 3f). While COSMIC Signature 1 has a relatively large contribution to mutation accumulation in the intestine in vivo, the contribution of this signature to the in vitro mutational pattern is limited. Likewise, the contribution of COSMIC Signature 5 to mutation accumulation in liver stem cells is more prominent in vivo than in vitro. Together, these findings further demonstrate that in vivo mutational processes only play minor roles in vitro, where other mutational processes are dominant. The in vitro-associated Signatures 8 and 18 are characterized by a large contribution of C > T changes, and Signature 18 has previously been linked to oxidative damage[32,35]. This is in line with our observation that C > T transversions are the dominant base substitutions in stem cells cultured under atmospheric oxygen levels. Hence, in vitro mutation accumulation seems to be mainly determined by culture-induced high levels of oxidative stress, resulting in largely similar mutational patterns, irrespective of the stem cell type, even though clustering analyses indicate that stem cell-type specific mutational processes are also active.

**The mutational risk of in vitro culture**. The in vitro mutational processes may lead to the transplantation of cells carrying pathogenic mutations. We therefore used the empirically defined stem cell-specific mutation rates, mutation spectra, and genomic distribution to model the risk of oncogenic mutations to occur during in vitro culture. This revealed a near-linear correlation between the cumulative number of pathogenic mutations in oncogenes[36] as a function of the number of stem cells that are generated in vitro (Fig. 4a). In liver ASCs, there is one oncogenic mutation per $3.5*10^6$ cells, in intestinal ASCs, one per $1.3 \times 10^7$ cells, and in PSCs one per $2.4 \times 10^7$ cells. When $10^8$ intestinal ASCs are produced in vitro (a high-end estimate required for a cellular transplantation), the chance for at least one oncogenic mutation in the population is approximately 1. To place this chance in perspective, we also calculated the number of oncogenic mutations that occur in the colon in vivo. The risk for $10^8$ in

vitro-produced intestinal ASCs is equivalent to the risk of accumulating an oncogenic mutation in any stem cell in human colon in vivo in 100 days (Fig. 4b). Because not all possible oncogenic mutations are relevant for all three stem cell types, we also looked at more specific mutations. For example, in human PSCs, dominant negative P53 mutations have been identified that confer a selective advantage to the cells in culture[19]. Based on our in vitro mutation accumulation results, we predict that these mutations occur once in every $\sim 2.0 \times 10^9$ PSCs (Fig. 4a). As another example, we focused on the $BRAF^{V600E}$ oncogenic mutation, which is found in ~10% of colorectal cancers[37]. In intestinal ASCs, this mutation occurs once in every $3.1 \times 10^{10}$ stem cells (Fig. 4a). The chance for having a $BRAF^{V600E}$ mutation in the total cell population is 0.018, when $10^8$ intestinal ASCs are produced in vitro. This chance is equivalent to the probability that a $BRAF^{V600E}$ mutation occurs in any colon stem cell in 112 days of adult age (Fig. 4c). Strikingly, the mutational risk for oncogenic mutations was not lower for cells cultured under reduced oxygen tension (Fig. 5a), even though the mutation rates were decreased (Fig. 2b). We reasoned that differences in the genomic distribution may form a possible explanation for this seemingly contradictory result. Mutations are depleted in PSCs cultured under 20% oxygen, but not in the PSCs cultured under 3% oxygen (Fig. 5b). Thus, the reduction in mutation rates under low-oxygen tension is largely outside genic regions, thereby having little effect on the oncogenic risk. On a cautionary note, in the low-oxygen group, the number of mutations in the coding sequence are rather low for genomic distribution analyses, and as a result, the risk assessment is less accurate.

## Discussion

Many studies have described genetic abnormalities in stem cells during in vitro culture and/or derivation. These findings seem to contrast with the observation that stem cells have a higher activity of DNA repair pathways, and repair damaged DNA more efficiently than differentiated cell types[38–40]. To systematically investigate genome stability of stem cells, and quantify mutation accumulation, we have applied WGS of PSCs and ASCs to identify genetic aberrations that were acquired during the culturing period in between two clonal steps. Some of the mutations measured in this study may have originated from the clonal steps and the accompanying enhanced cellular stress in the absence of cell–cell contacts and lack of autocrine and paracrine factors. A lengthy in vitro culture period was performed to maximize the contribution of mutations induced by routine culturing, and to minimize the contribution of mutations induced by the clonal step. The validity of this approach is clear from the mutational patterns of the cells grown under 3% and 20% oxygen. The mutational patterns are clearly different from one another, indicating minimal contribution of the mutations induced by the clonal steps.

An important distinction between our study and previous studies is that our approach enables to discriminate mutations that originate in culture from those that have an in vivo origin or arise as a result of the derivation process. For example, previously, we discovered that, in vivo, ASCs acquire approximately 40 mutations per year[24]. Due to the accumulation of mutations with age in vivo, ASCs of older individuals carry more mutations in vitro as well. By our experimental approach, we excluded these confounding factors, and demonstrate that the in vitro mutation rates for ASCs are nearly 40-fold higher than in vivo. For tumor-derived stem cells[41], mutation rates could be even higher due to pre-existing increased mutation rates in tumor cells. However, the in vivo number should be considered in a lifelong context

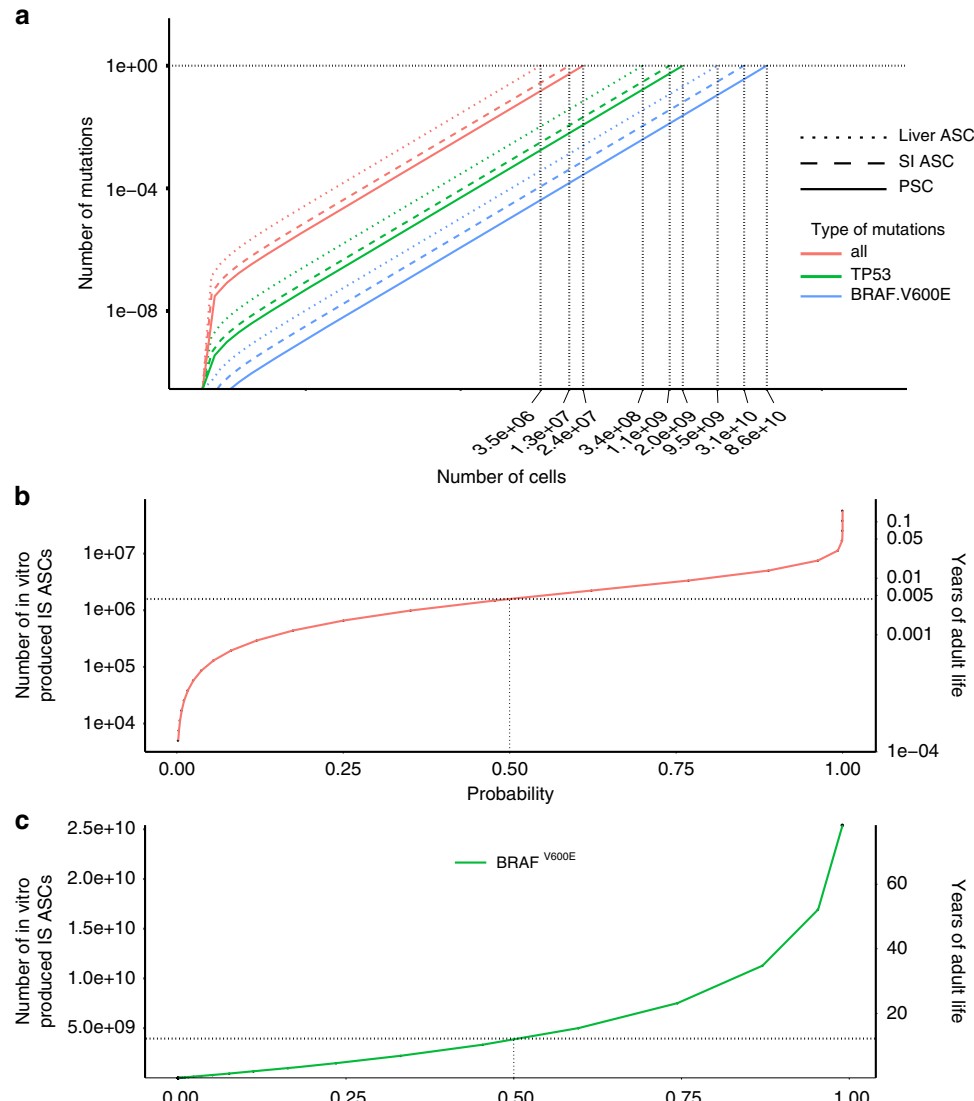

**Fig. 4 Modeling the oncogenic mutations in in vitro-cultured stem cells. a** The number of oncogenic mutations as a function of the number of in vitro-produced cells. **b** The probability of oncogenic mutations (*x*-axis) as a function of the number of in vitro-produced ASCs (primary *y*-axis,) and as a function of years of adult life (secondary *y*-axis). **c** The probability of *BRAF*^V600E mutations (*x*-axis) as a function of the number of in vitro-produced ASCs (primary *y*-axis), and as a function of years of adult life (secondary *y*-axis).

(decades) over a large population of stem cells, while the in vitro number relates to the duration of culturing (weeks or months) and the total number of cells required.

Our mutational signature analyses suggest that PSCs acquire fewer SBSs, indels, and SVs per population doubling than ASCs. This suggests that PSCs experience lower levels of DNA damage or have higher activity of DNA repair pathways. Previous studies have indeed demonstrated that human PSCs exhibit high activity of DNA repair pathways, leading to enhanced capacity to repair damaged DNA[40,42]. However, when taking into account the stem cell-specific mutation rates, mutation spectra, and genomic distribution, mathematical modeling indicates that PSCs acquire only slightly fewer oncogenic mutations than ASCs.

There is a strong correlation between the number of mutations and the number of cells produced. As a consequence, the majority of the mutations arise during the last population doubling, and are therefore present at very low frequencies, but also remain undetected in pool-based approaches. It is therefore impossible to rule out the risk that a population of cells that are being transplanted is devoid of any oncogenic mutation. This underscores the importance of accurate risk assessment. Our examples illustrate that oncogenic mutations will inevitably arise during culture as they do in vivo, but the risk for a specific oncogenic mutation, such as *BRAF*^V600E, is low. We therefore think that the observed in vitro mutation rates should not impede their future use in regenerative medicine. However, to avoid unnecessary mutation accumulation, we recommend to minimize the time in culture.

The in vitro mutation spectra of all three stem cell types are characterized by high numbers of C > T transversions, which have previously been linked to ROS. High ROS levels can cause oxidative damage to guanine, resulting in the formation of 8-oxoguanine. Incorporation of dAMP opposite 8-oxoguanine escapes exonucleolytic proofreading by DNA polymerases, generating C > T transversions during the next round of replication[31]. Previous studies demonstrated that human PSCs cultured under lower oxygen tension indeed experience less DNA damage[8,33]. For the first time, we demonstrate that lowering the levels of ROS in PSC cultures by

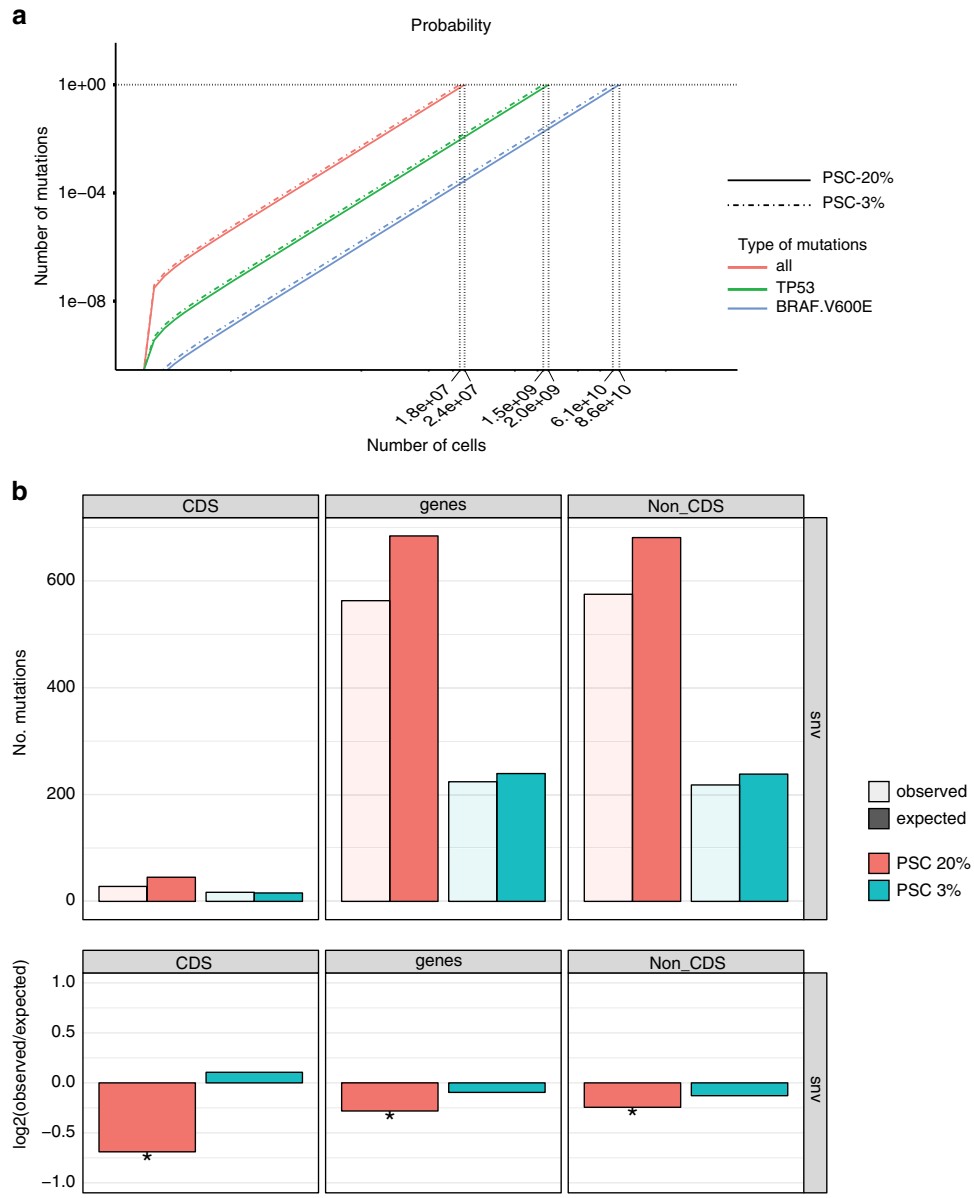

**Fig. 5 The effect of oxygen tension on the risk of acquiring oncogenic mutations in in vitro-cultured PSCs. a** The number of oncogenic mutations as a function of the number of in vitro-produced cells for pluripotent stem cells cultured under 20% and 3% oxygen. **b** Top panels: genomic distribution of the observed versus the expected amount of mutations per genome per doubling. Bottom panels: log₂ of the ratio between the total number of observed mutations versus the total number of the expected amount of mutations. Asterisks denote statistically significant differences between observed and expected (binomial test). CDS coding sequence.

reducing oxygen tension leads to a genome-wide reduction in the number of C > T changes, and a decrease in the total mutation rates. This is a clear proof of principle that the mutational burden can be influenced by tweaking culture conditions.

Taken together, the experimental approach of the current study provides a suitable framework for studies on the effects of mutagenic environmental factors during in vitro culture, and allows for further optimization of culture conditions to eventually reflect in vivo mutation rates, and to establish safe in human applications of these powerful cellular tools.

## Methods

**Stem cell culture**. All tissue culture products were from ThermoFisher Scientific, unless stated otherwise. Integration-free human induced PSCs were established from urinary cells using Sendai virus with ethical approval of the University Medical Center Utrecht (the Netherlands) under study ID NL55260.041.15 15-736/M. Primary cells were collected by centrifugation of the urine of two healthy anonymous

male research participants who were not involved in this study, and who have provided explicit informed consent for ips cell derivation and genetic characterization[43]. The cell pellets were washed with PBS supplemented with antibiotics. The cell pellets were resuspended in primary medium containing DMEM/high glucose and Ham's F12 nutrient mix, supplemented with 10% (vol/vol) fetal bovine serum, pen/strep, and Renal Cell Growth Medium SingleQuots™ (Lonza) 2.5 µg ml⁻¹ ampho-tericin B. The cells were plated onto gelatin-coated plates and incubated at 37 °C in a humidified atmosphere and 5% $CO_2$. The medium was refreshed daily. Ninety-six hours after plating, the medium was switched to renal cell growth medium sup-plemented with the SingleQuots™. At near-confluency, the cells were split into new gelatin-coated plates using TrypLE Express. Urinary cells were subsequently repro-grammed by transduction with Sendai virus carrying human *KLF4*, *OCT4*, *SOX2*, and *C-MYC* using the CytoTune™ iPS 2.0 Sendai Reprogramming Kit according to the manufacturer's protocol. Approximately, 10 days after transduction, individual iPS colonies were manually picked and further expanded. The iPS cell lines and the human embryonic stem cell line H9 were cultured in E8 medium on tissue culture plates coated with Geltrex (ThermoFisher) or Matrigel (Corning)[44]. RNA-seq ana-lysis confirmed that the iPS cells closely resembled human embryonic stem cells (Fig. 6). Clonal steps were performed by limiting dilution of a single-cell suspension in 96-well tissue culture plates coated with Geltrex or Matrigel. To enhance cell

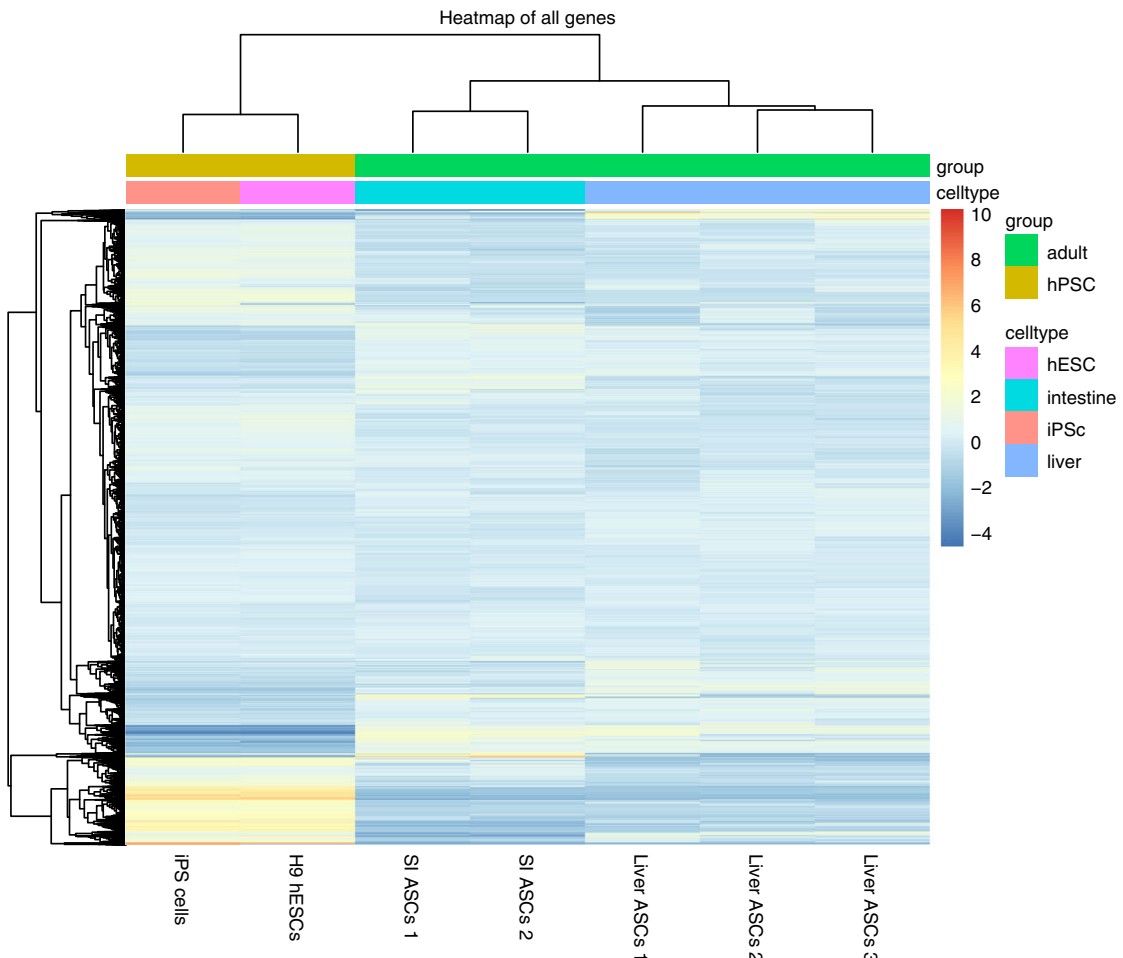

**Fig. 6 RNA-seq analysis of human adult and pluripotent stem cells.** Heatmap and unsupervised hierarchical clustering of RNA-seq data for all genes for human PSCs, human ES cells, human liver stem cells, and human intestinal stem cells.

survival after the clonal steps of the PSCs, E8 medium was supplemented with RevitaCell™. In between two clonal steps, the PSCs were cultured for 3 months to accumulate mutations. After a 3-month culture, a second clonal step was performed. The resulting clones were expanded until enough material was produced for WGS. To filter out germline variants, we used a cell pellet from the preclonal bulk culture as reference sample.

For ASCs, we used existing human liver and intestinal stem cell lines[24,45]. For these lines, the patients' informed consent was obtained, and the use for research purposes was approved by the ethical committees of University Medical Center Utrecht and Erasmus Medical Center, Rotterdam. Liver stem cell lines were cultured in Advanced DMEM/F12 with HEPES, Glutamax, and Penicillin–Streptomycin. This medium was further supplemented with N2, B27, 1.25 mM N-Acetylcysteine (Sigma), 10 nM gastrin (Tocris), Primocin (Invivogen), and the following growth factors: 50 ng per ml EGF (Peprotech), 10% RSPO1-conditioned media (homemade), 100 ng per ml FGF10 (Peprotech), 25 ng per ml HGF (Peprotech), 10 mM Nicotinamide (Sigma), 5 μM A83.01 (Tocris), and 10 μM FSK (Tocris). For the first 3 days after each clonal step, the medium contained 25 ng per ml Noggin (Peprotech), 30% Wnt3a-conditioned medium (homemade)[46], 10 μM Y27632 (Abmole), and stem cell cloning recovery solution (Stemgent). Intestinal stem cell lines were cultured in Advanced DMEM/F12 with HEPES, Glutamax, and Penicillin–Streptomycin. This medium was further supplemented with 50% Wnt3a-conditioned medium, 20% R-Spondin-conditioned medium, B27, 1.25 mM N-Acetylcysteine, 10 mM Nicotinamide, Primocin, 0.5 mM A83-01, 30 mM SB202190 (Sigma), recombinant human Noggin, and 50 ng per ml recombinant human EGF. For the first 3 days after each clonal step, the intestinal organoid medium was supplemented with 10 μM Y27632.

To make clonal stem cell lines, organoids were enzymatically dissociated, and the resulting single-cell suspension was FACS-sorted to remove all doublets. The single-cell suspension was resuspended in Matrigel or BME (Pathclear) for intestinal and liver stem cells, respectively, and plated as a limiting dilution series. After 2–3 weeks, individual organoids were manually picked and mechanically fragmented and plated under regular culture conditions. Clonal lines were further cultured for another 53–146 days after which a second clonal step was performed. The resulting subclones were expanded until enough material was available for

WGS. Per stem cell line, a blood sample of the same donor was also sequenced to enable filtering for germline variants.

**DNA/RNA isolation, library prep, and sequencing.** DNA was isolated manually using the genomic tip 20-G kit (Qiagen) or automated using the Qiasymphony (Qiagen). From 200 ng of genomic DNA, DNA libraries were generated for Illumina sequencing using standard protocols (Illumina). Libraries were sequenced 2 × 150 bp paired-end to 30× base coverage with Illumina HiSeq Xten. For RNA sequencing, human iPS cells, ES cells, liver ASCs, and intestinal ASCs were collected in Trizol and subjected to total RNA isolation with the QiaSymphony SP using the QiaSymphony RNA kit (Qiagen, 931636). Subsequently, 50 ng of total RNA was used to prepare mRNA-sequencing libraries using the Illumina Neoprep TruSeq stranded mRNA library prep kit (Illumina, NP-202-1001), followed by paired-end sequencing (2 × 75 bp) of the RNA libraries on a Nextseq500 to >20 million reads per sample. All sequencing data have been deposited at the European Genome-phenome Archive (http://www.ebi.ac.uk/ega/) under accession numbers EGAS00001002955, EGAS00001000881, and EGAS00001001682.

**Sequencing analysis.** RNA-sequencing reads were mapped to the human reference genome GRCh37 with STAR v.2.4.2a[47], and the BAM files were indexed using Sambamba v0.5.8. Reads were counted using HTSeq-count 0.6.1p1, and read counts were normalized using DESeq v1.28.0. Nonsupervised hierarchical clustering was performed using DESeq2.0 v1.20. DNA-sequencing reads were mapped against human reference genome GRCh37 using Burrows–Wheeler Alignment (BWA–MEM) v0.7.5a.

To determine the callable loci region of a sample, we used the CallableLoci tool from GATK (version 3.4.46) and considered genomic regions with a minBaseQuality score of 10, a minMappingQuality score of 10, a minDepth of 20, and a minDepthForLowMAPQ of 20. Next, we intersected the callable loci regions of the bulk, clonal, and subclonal sample, and only considered SBS and indel variants located in these regions. For each sample, all mutations were normalized to the callable genome to determine the total number of mutations per whole genome (Supplementary Data 1). To identify all SBSs of the clonal cultures, we used our

data analysis pipeline that enables the identification of somatic SBSs with a confirmation rate of ~91%[24]. In short, we performed basic protocols 1–2 of the Genome Analysis Toolkit (GATK) best-practices workflow for germline single-nucleotide polymorphisms (SNPs) and indels in whole genomes to identify SBSs[48]. We subsequently generated a catalog of high-quality in vitro-induced SBSs using a custom Single Nucleotide Variant Filtering pipeline (SNVFI available at https://github.com/UMCUGenetics/SNVFI). The SBS call set is further filtered on the basis of quality parameters (base call quality >100, minimum coverage at variant position >20, and VAF-filtering threshold of 0.3[21]) and dbSNP v137.b37[49]. In addition, positions that were found to be variable in at least three unrelated individuals were excluded as these represent either unknown SNPs or recurring sequencing and/or calling artifacts[21]. Furthermore, events with any evidence in the reference sample (alternative depth > 0) were excluded. Clonality of the cultures was verified by a distribution around 0.5 of the VAFs of the mutations. By filtering against all mutations with allele frequencies below 0.3, we excluded all mutations that arose during the culture period after the clonal steps. To identify all the mutations that originated in the culture period in between both clonal steps, the mutations identified in the second clonal culture were filtered for those present in the bulk and the first clonal culture. The resulting number of SBSs was divided by the population doublings, to obtain the point mutation rate for each stem cell type. An ANOVA was performed to determine whether the mutation rates differ significantly between the stem cell types, and a two-sided Student's $t$ test was performed to test for significance differences in mutation rates between 20% and 3% oxygen conditions. Mutational pattern analysis was performed with the MutationalPatterns R-package[50]. Seven-type mutation spectra were extracted from the VCF files. Subsequently, the 96-type mutational profiles were generated, and the average profile (centroid) was determined for the assessed SC types. The COSMIC mutational signatures were downloaded from COSMIC website (https://cancer.sanger.ac.uk/cosmic). Centroids were reconstructed with the 30 COSMIC mutational signatures. Subsequently, signatures that contributed at least 10% to one or more centroids were selected. Cosine similarities between samples and/or signatures and relative contributions of signatures were calculated using MutationalPatterns. Gene definitions for hg19 were retrieved from the University of California, Santa Cruz (UCSC) Genome Browser as a TxDb annotation package from Bioconductor. Epigenetic status datasets were downloaded from the ENCODE website (https://www.encodeproject.org/) as BED files.

Functional consequences of coding mutations were identified using the VariantAnnotation R-package. To test for enrichment and depletion of SBS within genes, we used a one-sided binomial test with MutationalPatterns[50]. To assess the presence of the mutations within genes and to predict their effect, the SBSs were annotated using SnpEff[51].

Unfiltered indel catalogs were extracted from the germline GATK calls acquired in "Single nucleotide variants". We only considered indels within the callable autosomal genome of the clone, subclone, and control sample. We subsequently excluded indels that are present in the dbSNP database to filter out known germline variants, as well as those that are present in any other sample used in this study, to exclude rare germline variants and somatic calling artifacts. From these, we only considered indels with a filter "PASS" from VariantFiltration, with a GATK genotype quality of at least 99, and those with a REF and ALT read support between 10× and 60× in clone, subclone, and control sample (average read depth is 30×). Subsequently, we followed the same approach as with SBSs to detect the culture-associated variants. Here, indels from the subclonal sample with any evidence in the respective control and clonal sample (with VAF > 0.3) were filtered out. From these, only clonal indels with a VAF support greater than 0.3 were considered.

We extrapolated the number of indels to the human autosomal genome. Subsequently, the number of indels was divided by the number of population doublings, to obtain the indel mutation rate for each stem cell type. An ANOVA was performed to determine whether indel mutation rates differ significantly between stem cell types. Finally, we used SnpEff variant annotation to predict the effect of the mutations on the genes[51].

**Modeling of in vitro mutation accumulation.** To calculate population-doubling time, we used the split ratios to calculate the number of cells we would have obtained under maximum expansion after a culture period of 44 days. Next, we applied

$$D = \frac{T \cdot ln(2)}{ln(Xe/Xb)}, \quad (1)$$

where $T$ is the incubation time in any unit, $Xb$ is the cell number at the beginning of the incubation time, and $Xe$ is the cell number at the end of the incubation time. To calculate the number of mutations in the protein-coding fraction of the genome, we applied

$$M(t) = 0.015 \cdot dp \cdot \mu \cdot N_0 \cdot 2^{(mt/D) \cdot G(t)},$$

where $M(t)$ is the number of mutations at timepoint $t$, 0.015 is the coding fraction of the whole genome and $dp$ is the degree of depletion in the CDS, $\mu$ is the number of mutations per cell cycle, $N_{(0)}$ is the number of initial cells, $mt$ is the cell cycle

length, and $G(t)$ is the number of generations at timepoint $t$. For ASCs, we used a cell cycle length of ~26 h, and for PSCs, we applied a cell cycle length of ~18 h[52].

We used a list of oncogenic mutations in driver genes[36] to calculate the number of mutations in the coding sequence that activate cancer driver genes. For details on how this list was compiled, we refer to Tamborero et al.[36]. The probability $P$ that mutation types ($C \rightarrow A, C \rightarrow G, C \rightarrow T, T \rightarrow A, T \rightarrow C,$ and $T \rightarrow G$) happen in the genome can be derived from the mutation spectrum of the cell type. To calculate the number of oncogenic mutations as a function of the number of cells, we applied

$$M^{\text{active}}(N) = 0.015 \cdot dp \cdot \mu \cdot N \cdot \sum_{\substack{X \epsilon \{C, T\} \\ Y \epsilon \{A, C, G, T\} \\ X \neq Y}} \left( P_{X \rightarrow Y} \cdot \frac{n_{X \rightarrow Y}}{L} \right), \quad (2)$$

where $M^{\text{active}}$ is number of mutations that activate driver genes, $dp$ is depletion in CDS, $\mu$ is the mutation rate, $N$ is the number of cells, $P_{X > Y}$ is the chance on $X > Y$ mutation based on the mutation spectrum, $n_{X > Y}$ is the number of positions where $X > Y$ mutation results in oncogene activation, and $L$ is the length of CDS.

To calculate the probability for an oncogenic mutation as a function of the number of cells, we applied the following formula, where $Z$ is the number of activating mutations

$$\mathbb{P}(Z \geq 1) = 1 - \mathbb{P}(Z = 0)$$

$$= 1 - \left( \prod_{\substack{X \epsilon \{C, T\} \\ Y \epsilon \{A, C, G, T\} \\ X \neq Y}} \left( \frac{L - n_{X \rightarrow Y}}{L} \right)^{P_{X \rightarrow Y}} \right)^{0.015 \cdot dp \cdot \mu \cdot N} . \quad (3)$$

To calculate the probability for an oncogenic mutation in vivo in $10^8$ colon stem cells as a function of the number of years ($t$) in adult life, we applied the following formula:

$$\mathbb{P}(Z \geq 1) = 1 - \mathbb{P}(Z = 0)$$

$$= 1 - \left( \prod_{\substack{X \epsilon \{C, T\} \\ Y \epsilon \{A, C, G, T\} \\ X \neq Y}} \left( \frac{L - n_{X \rightarrow Y}}{L} \right)^{P_{X \rightarrow Y}} \right)^{0.015 \cdot dp \cdot 10^8 \cdot 40 \cdot t} . \quad (4)$$

**Reporting summary**. Further information on research design is available in the Nature Research Reporting Summary linked to this article.

## Data availability

The source-sequencing data underlying Fig. 1c–e and Figs. 2–4 is available at the European Genome-phenome Archive through (http://www.ebi.ac.uk/ega/) under accession numbers EGAS00001002955, EGAS00001000881, and EGAS00001001682. All the other data supporting the findings of this study are available within the article and its Supplementary information files and from the corresponding author upon reasonable request. Publicly available data were also used from dbSNP (dbSNP v137.b37) and ENCODE (https://www.encodeproject.org/).

## Code availability

Filtered vcf files and supporting code are available at https://github.com/UMCUGenetics/in_vitro_mutational_load. Mutational pattern analyses were performed with the MutationalPatterns R-package, available at https://bioconductor.org/packages/3.6/bioc/html/MutationalPatterns.html. Custom script to model driver probabilities is available at https://github.com/bastiaanvdroest/DriverProbabilities.

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

## Acknowledgements

The authors would like to thank Luc J. W. van der Laan of the Erasmus MC for providing liver samples and Sabine Middendorp for providing intestinal samples, for which previously published data have been used in the current study. The authors thank the USEQ for sequencing support and the UBEC for bioinformatical support. This work was financially supported by the NWO Gravitation Program Cancer Genomics.nl and the NWO/ZonMW Zenith project 93512003 to E.C.

## Author contributions

M.J., E.C., and E.K. wrote the paper. N.B., R.v.B., M.J., J.K., and E.K. performed wet-lab experiments. M.J., R.J., A.V.H., M.L., and E.K. performed the bioinformatical analyses. R.J., A.V.H., B.vd.R., and E.K. performed the mathematical modeling. S.B. provided the bioinformatical support. R.v.B., E.C., M.J., and E.K. were involved in the conceptual design of the study. E.C. supervised the study.

## Competing interests

The authors declare no competing interests.
