## [Peer Review File · Nature Communications]

Reviewers' comments:

Reviewer #1 (Remarks to the Author):

Kuijk et al. quantified and characterized genetic aberrations in human iPSCs and adult stem cells during in vitro cell culture. They further compared in vitro and in vivo mutation data, revealing several novel and important insights. Detailed analysis of genetic aberration in cultured iPSCs in direct comparison with other stem cells is an important step towards the medical application of these stem cells. The manuscript is well written, and the topic is also timely. However, the authors need to consider the following major remarks.

Major points

#1. My major concern in this study is that the authors only use one iPSC cell line which they established. The authors conclude that mutation frequency in iPSCs is about half of those in ASCs (without even "biological" replicates for the 1st clonal step). One of the major issues of iPSCs is the genetic and epigenetic variations as reviewed by Liang and Zhang in *Cell Stem Cells* (2013) doi: [10.1016/j.stem.2013.07.001]. Although the similarities between their iPSCs and H9 hESCs are presented in Supplemental figure 6, the reviewer think it is important to include several iPSC cell lines (and optionally human ESCs), and to compute solid mutation rate of iPSCs. Awing to the sharp drop of sequencing cost in recent years, some more experiments would not be cost prohibitive.

#2. The authors show that in vitro mutation signature is characterized by C to A changes, which are linked to oxidative stress conditions. As the authors discuss, lowering the levels of ROS in culture may be a way to reduce the number of mutations. It would be ideal to provide experimental data to support this for the readers of this journal.

Minor points

#1. Color code in Figure 1d and 1f. The color code for observed and expected seems only valid for iPSC but not for SI and Liver. The abbreviation for SI (spelled S.I. in Figure 2) can also be shown in the figure legend.

Reviewer #2 (Remarks to the Author):

In their study "Mutational impact of culturing human pluripotent and adult stem cells" Kuijk et al. describe the comparative analysis of human induced pluripotent stem cells (iPS cells) and adult stem cells (ASCs) from intestine and liver for accumulation of de novo mutations during in vitro culture with in vivo mutations by whole genome sequencing.

The major findings include:

- 1) iPS cells, intestinal ASCs and liver ASCs have different mutation rates and accumulated 3.5 ± 0.5 , 7.2 ± 1.0 and 8.4 ± 3.6 base substitutions per population doubling, respectively.
- 2) The annual in vitro mutation accumulation rate of ASCs is ~40- fold higher than the in vivo rate of ~40 base pair substitutions per year.
- 3) Irrespective of stem cell type, mutational signatures in vitro are distinct from the in vivo mutational signature. This in vitro signature is characterized by C to A changes that have previously been linked to oxidative stress conditions.
- 4) Stem cell-specific mutational signatures and differences in transcriptional strand bias were observed, indicating differential activity of DNA repair mechanisms between stem cell types in culture.

Also, the authors claim that their approach, in contrast to previous studies, is able to discriminate mutations that originate in culture from those that have an in vivo origin or arise as a result of the derivation process (see page 5, discussion).

In general, the bioinformatics filtering and analyses appear appropriate, however the paper is difficult to read as comprehensive explanations and definitions are missing.

General questions / comments on the methodology:

Suppl Figure 1 is important to understand the study and should become main Fig. 1. However, figure and corresponding legend are not comprehensive enough and should be more detailed, or it should at least be referred to sections of the manuscript where relevant details are provided.

In particular following aspects that are important to judge the experimental design and results do not become clear in the paper:

- 1) As I understood, genomic DNA from urothel bulk culture of the corresponding donor in case prior to reprogramming was used as reference sample in case of iPSCs, correct? How many population doublings did happen in the primary culture of urinary epithelium between isolation until reprogramming and picking of single cell iPSC clones? This has to be described more clearly.
- 2) In case of liver and intestinal stem cell lines, blood of the corresponding donor was used as reference sample, correct? It would be better to use an intestinal or liver biopsy as reference sample (with the disadvantage that also other cell types than stem cells are included) or alternatively the bulk population that was used for the 1st cloning step as reference!
- 3) If blood has been used as reference sample, it has to be considered that somatic mutations in the sampled white blood cells may be very different than in the cell population utilized for isolation of intestinal stem cells and hepatic stem cells. Besides germ line mutations there are somatic mutations in individual cell clones that undergo a positive or negative selection pressure dependent on the affected cell lineage. Therefore it may be that certain mutations are rare (and undetectable by direct WGS) in blood cells, but become enriched in intestinal or livers stem cells.
- 4) The authors should analyse and present a list of mutations that were not detectable in the parental cells but were detectable in the first clonal culture (representing de novo mutations during reprogramming and/or initial culture phase, or rare mutations inherited by the analysed clones from their parental cells).
- 5) How many population doublings occur for the different cell types between first single cell cloning until sequencing of the clonal bulk culture? How many population doublings occur until the second cloning step for each cell type (53-146 days)? This should be included in supplementary table 1.
- 6) The bulk population after the extended in vitro culture period (used for the 2nd cloning step) should also be sequenced and results should be compared to the sequencing results of the early bulk culture after 1st cloning and the bulk culture after 2nd cloning. This will allow to discriminate between de novo mutations that occurred during normal culture and mutations that occur or become selected due to the very stressful conditions during single cell cloning.
- 7) In the Material and Methods section it is mentioned that "Welch Two Sample t-tests were performed to determine whether the mutation rates differ significantly between the stem cell types". To my knowledge, more samples (in particular more than 3 iPSC samples) are required to use a T-test for analysis. In the results section, only the ANOVA test is mentioned. Which test has finally been used? The authors should comment on the choice and applicability of the applied tests.
- 8) Number of three analysed iPSC subclones from only one donor may be not sufficient to be representative. At least three additional iPSC subclones from another primary clone should be included. Are the four liver and the six intestinal subclones each derived from one primary clone?

Results:

Page 3: "We found a population doubling time of ~23h for human iPSCs which is approximately twice as fast as for human liver and intestinal ASCs, that have population doubling times of ~46h and ~44h, respectively (figure 1a)." Was the level of cell death determined and taken into account for this calculation?

Page 3: "Base pair substitutions were significantly depleted in genic regions for all three stem cell types (figure 1d)." What means „depleted“? Less than statistically expected? What is the basis for this calculation?

Page 3: "None of the non-synonymous mutations affected known cancer genes based on the census of human cancer genes²¹ or have previously been described to confer a selective advantage over stem cells in culture^{6,14}. We are therefore confident that the observed mutations provide an unbiased insight into mutation accumulation free of selection." Lack of description as cancer gene does not necessarily mean that there is no growth advantage for a specific mutation. Also there are not many small scale mutations described so far that provide a selective advantage during culture. Therefore, it cannot be concluded that there is an "unbiased insight into mutation accumulation free of selection". In addition, it may well be that a certain mutation does not provide any selection during normal culture but during single cell cloning, for instance mutations in apoptosis-related genes.

Discussion section:

The authors claim that "their approach, in contrast to previous studies, is able to discriminate mutations that originate in culture from those that have an in vivo origin or arise as a result of the derivation process" (see page 5, discussion).

If the WGS data of the clonal cultures are compared with corresponding nonclonal reference sample (bulk of urothelial cells that were parental cells of the analysed iPSCs, and corresponding donor blood in case of ASCs) it is of course possible to exclude germline variants and variants that are present in the reference sample at relatively high levels (higher than detection limit of the direct WGS, which depends on the error rate of the applied Illumina sequencing approach (sequencing-by-synthesis). What is the detection level? 10%? Or even higher?

Importantly, direct WGS excludes most mutations that have been acquired over a lifetime after the first cell doublings in the early embryo and that did not become enriched in urothelial cells or blood cells during life.

It has to be considered that the parental cell population, which has been used for reprogramming of iPSCs or isolation of ASCs, represents a genetic mosaic. Probably every cell has a unique signature of mutations that had been acquired over lifetime. Therefore, both, mutations that are too rare in the reference cell population but are present in the specific cell clone, and mutations that occur during the first culture period and the first cloning step, will be detectable in the bulk culture after the first cloning step. Therefore, it is not always possible to discriminate between mutations that originate in culture and those that have an in vivo origin or arise as a result of the derivation process

Materials and Methods section:

Page 9: Sequencing and Bioinformatics: How are "callable regions" specifically defined? Which minimal coverage?

Page 9: "To identify all SNVs of the clonal cultures we used our data analysis pipeline that enables the identification of somatic SNVs with a confirmation rate of ~91%¹⁸." The authors should provide more

information. Why 91%?

Page 9: "The SNV call set is further filtered on the basis of several quality parameters and dbSNP v137.b37.40." Also here more information is required!

Page 9: "Welch Two Sample t-tests were performed to determine whether the mutation rates differ significantly between the stem cell types." I guess T-test is not appropriate if only 3 or 4 samples are available.

Page 9: "We subsequently excluded indels that overlap with a SNV, indels that have a dbSNP ID and no COSMIC ID, and indels that are present on a blacklist of 3 unrelated samples (BED file available upon request)." What is this black list?

Page 10: CNVs were called using Control-FREEC. What is the minimal coverage for those analyses? What fraction of the total genome matched this minimal coverage?

Figure & Legends:

Legends should always be placed below the corresponding figure, otherwise it is confusing.

Legend Fig 1: Should termed "Single Nucleotide Variants" instead of "mutations"; 1e) more different colours should be used for "Coding" and "non-synonymous"; it would be helpful for many readers to have a definition of (non)synonymous included in the legend f) How is promoter-flanking defined? Everything that is not in promoters? Certain distance around promoters?

Conclusion:

I think that the topic of acquired mutations in iPSCs and ASCs is still a very important topic, which has particular relevance for cellular therapies. Clearly, many important questions in this field remained unanswered yet.

The study of Kuijk et al. provides novel data, but suffers from methodological limitations and also partially by conclusions that cannot be drawn from the data. For publication in Nat Communications, definitely major revisions would be required.

Reviewer #3 (Remarks to the Author):

In this manuscript, Kuijk et al. analyze the burden and spectrum of mutations that arise during the in vitro culture of human pluripotent and adult stem cells. By applying an elegant study design that includes two clonal steps, the authors were able to capture and analyze the mutations that arose during a defined period of time (and a defined number of cell divisions) in subclones of three stem cell types: iPSCs, liver ASCs and intestinal ASCs. Their main findings are that the in vitro mutation rates for ASCs are ~40-fold higher than those reported in these cells in vivo; that functionally-important genomic regions are less commonly altered in all three stem cells types (as expected); and that the in vitro mutational processes are different from the in vivo mutational processes, and are characterized by C to A changes that have been previously linked to oxidative stress.

This is an interesting study that takes an important step towards understanding the mutational

processes that affect cultured stem cells. Such understanding is necessary to improve stem cell culture practices and ultimately enable safer use of these cells in regenerative medicine. Overall, the experiments and analyses are well-designed and carefully-performed. However, it is not clear how generalizable the findings of this study really are. A major limitation is that a single parental population was used to derive a few subclones of each stem cell type, so that eventually comparisons were performed across clonal populations that originated from a single iPSC line, a single liver ASC line and a single intestinal ASC line (i.e., n=1 per SC type). To what extent these cell lines represent their stem cell “type” is not clear, and cannot be addressed without further experiments. It would be much more informative to compare clonal cultures derived from independent parental populations (taken from different individuals) in order to be able to assess the variability that exists within each cell type.

The manuscript could also benefit from some additional elaborations and clarifications, as described below:

1. The authors argue that their findings “demonstrate that in vivo mutational processes only play minor roles in in vitro mutation accumulation”. This is surprising and interesting, and the manuscript clearly provides evidence for in vitro mutational processes that are different from the in vivo ones. But if the role of the in vivo mutational processes is only minimal, why does hierarchical clustering based on the relative contribution of COSMIC signatures (Fig. 2c) show the in vitro ASCs to cluster together with the in vivo liver ASCs and apart from the other in vitro groups? Can this be better clarified?
2. The authors state that “none of the non-synonymous mutations affected known cancer genes based on the census of human cancer genes, or have previously been described to confer a selective advantage over stem cells in culture. We are therefore confident that the observed mutations provide an unbiased insight into mutation accumulation free of selection.” I do not believe that we can be so confident that selection is not involved. First, the list of genes involved in cancer that was used for the comparison (ref 21) is a census of amplified and overexpressed human cancer genes, and is thus partial by definition (it does not cover tumor suppressors, for example). Second, the selection pressures in culture are likely to be at least somewhat different from those experienced in vivo, and somewhat tissue-type specific; however, the list of genes selected for in stem cell cultures, which was used for comparison here, is limited to pluripotent stem cells (refs 6 and 14). Can the authors compare their mutated gene list to additional/extended gene lists? This statement should also be toned down
3. What specific comparisons were performed in Fig. 1b,c? The legend says ANOVA, but multiple p-values appear (Fig. 1b) and the p-values are presented above specific bars (Fig. 1b,c). Are these p-values from contrast analyses? Which contrasts do the shown asterisks represent? This should be clarified in the Figures and in the Methods.
4. Given that each stem cell type is cultured under distinct culture conditions, it is impossible to tease apart the role of the cell lineage and that of the culture conditions in shaping the mutational landscapes of the cultured cells. Can the authors comment on this point? The authors keep referring to “stem cell-specific” mutational processes but these could really be “culture conditions-specific” mutational processes. Can the authors comment on that?
5. One of the most interesting findings of the study – and certainly the finding with most practical implications – is that all cultured stem cells are enriched for C to A changes, which may be linked to oxidative stress. However, the study does not provide any direct evidence that oxidative stress indeed plays a major role in the mutational processes. An experimental demonstration that reducing ROS

levels could change the rate and the spectrum of culture-acquired mutations (even if provided only for one of the SC types) would significantly increase the impact of this work.

Reviewer #1 (Remarks to the Author):

Major points

#1. My major concern in this study is that the authors only use one iPSC cell line which they established. The authors conclude that mutation frequency in iPSCs is about half of those in ASCs (without even “biological” replicates for the 1st clonal step). One of the major issues of iPSCs is the genetic and epigenetic variations as reviewed by Liang and Zhang in Cell Stem Cells (2013) doi: [10.1016/j.stem.2013.07.001]. Although the similarities between their iPSCs and H9 hESCs are presented in Supplemental figure 6, the reviewer think it is important to include several iPSC cell lines (and optionally human ESCs), and to compute solid mutation rate of iPSCs. Awing to the sharp drop of sequencing cost in recent years, some more experiments would not be cost prohibitive.

We agree with the reviewer that biological replicates would make our findings much more interesting. Therefore, we have included an additional iPSC cell line from a different donor and the H9 ES cell line. For both pluripotent stem cell lines we generated clonal lines. Clones were cultured for 3 months to allow mutations to accumulate, followed by a 2nd clonal step to identify all the genetic variants that have been accumulated in the individual cells that gave rise to these subclones. Together with the previous results, this adds up to *in vitro* mutation accumulation data of 5 subclones from three genetically distinct pluripotent stem cell lines. In spite of their different origins (the iPSC cells were derived from renal cells, whereas the ES cell line is of course of embryonic origin), the variation between these lines in mutation rates and mutational patterns is very low. The additional samples lend further support to our finding that in culture pluripotent stem cells acquire fewer mutations per genome per doubling than ASCs. However, in our opinion, a more important message of the manuscript is that in spite of their different origins (ASCs from the liver, ASCs from intestine, iPSC cells from renal cells, and the ES cells of embryonic origin), all show the *in vitro* signature (e.g. many C to A changes). In our extensively revised manuscript, we have now emphasized the finding that *in vitro* culture induces specific mutation types irrespective of cell type and we place less emphasis on the differences between the different stem cell types.

#2. The authors show that in vitro mutation signature is characterized by C to A changes, which are linked to oxidative stress conditions. As the authors discuss, lowering the levels of ROS in culture may be a way to reduce the number of mutations. It would be ideal to provide experimental data to support this for the readers of this journal.

To confirm that the C>A changes are caused by oxidative stress, we have also cultured clones of the three pluripotent stem cell lines for three months under low oxygen levels (3%), followed by a second clonal step to measure all the genetic variants in the individual cells that gave rise to the subclones. Mutation frequencies under low oxygen conditions were significantly lower than under high oxygen levels (20%). In addition, the mutation spectra of cells cultured under low oxygen conditions were significantly different from those cultured under high oxygen conditions. In particular, the relative number of C>A were greatly reduced, whereas the relative number of C>T and C>T at CpG sites were enhanced. These findings strengthen our conclusion that *in vitro* culture of stem cells induces mutation accumulation through oxidative stress.

Minor points

#1. Color code in Figure 1d and 1f. The color code for observed and expected seems only valid for iPS but not for SI and Liver. The abbreviation for SI (spelled S.I. in Figure 2) can also be shown in the figure legend.

All figures have been updated, including the color codes. The abbreviation has been changed to SI ASC

Reviewer #2 (Remarks to the Author):

Suppl Figure 1 is important to understand the study and should become main Fig. 1. However, figure and corresponding legend are not comprehensive enough and should be more detailed, or it should at least be referred to sections of the manuscript where relevant details are provided.

We thank the reviewer for this suggestion and we have adapted the manuscript accordingly. Suppl Figure 1 is now figure 1a and more details are provided in the figure legend.

In particular following aspects that are important to judge the experimental design and results do not become clear in the paper: 1) As I understood, genomic DNA from urothel bulk culture of the corresponding donor in case prior to reprogramming was used as reference sample in case of iPSCs, correct? How many population doublings did happen in the primary culture of urinary epithelium between isolation until reprogramming and picking of single cell iPSC clones? This has to be described more clearly.

We have used early passage cells for reprogramming (within 3 passages) and the first iPS cell colonies were picked within 10 days after adding the virus. However, this is not very relevant for our manuscript. Our experimental design is specifically set up to measure the mutations that accumulate in between the two clonal steps. All germline variants are filtered using the reference sample. The mutations that have been acquired during development, life, cell line derivation (reprogramming), and subsequent culture until the first clonal step are filtered out using the WGS data of the clone. Mutations that have a VAF < 0.3 in the clone are not filtered from the data of the subclone, as these likely occurred after the clonal step. We apologize for the unclarity, and have changed the legend to the figure (1a) where we explain the experimental set-up as well as the first paragraph of the results section to better clarify our experimental design.

2) In case of liver and intestinal stem cell lines, blood of the corresponding donor was used as reference sample, correct? It would be better to use an intestinal or liver biopsy as reference sample (with the disadvantage that also other cell types than stem cells are included) or alternatively the bulk population that was used for the 1st cloning step as reference!

We disagree with the reviewer that it would be better to use a biopsy as a reference sample. However, the reviewer is right in saying that the 1st clone is an important reference for the 2nd clone, which is why we chose this experimental setup in which we use the first clone indirectly as a reference for the second clone. Briefly, we first determine all variants that are in the clone, by subtracting all germline variants from the bulk. Next we determine all variants that are in the subclone, also by subtracting all germline variants from the bulk. The subclone will have exactly the same variants as the clone and the additional variants that have been acquired during the *in vitro* culture period. After subtraction of all clonal variants, we end up with all variants that are unique to the subclone. What sample is being used as a reference is not very important, because the germline variants will be present both in bulk and in blood. If bulk is used as a reference, there might be bulk specific variants that are absent in blood. These variants will be considered germline (while they are not) and therefore these will be subtracted from both the clone and subclone. If, on the other hand, blood is used as a reference, bulk-specific variants will be detected as variants both in the clone and the subclone. These variants will be subtracted from the subclone, together with any other variants that are present in the clone but not the blood reference sample. Thus, whichever reference is used (blood or bulk), the end result (list of variants unique to the subclone) is the same. As mentioned in the previous comment, we apologize for the unclarity, and have amended figure (1a) as the first paragraph of the results section to better clarify our experimental setup.

3) If blood has been used as reference sample, it has to be considered that somatic mutations in the sampled white blood cells may be very different than in the cell population utilized for isolation of intestinal stem cells and hepatic stem cells. Besides germ line mutations there are somatic mutations in individual cell clones that undergo a positive or negative selection pressure dependent on the affected cell lineage. Therefore it may be that certain mutations are rare (and undetectable by direct WGS) in blood cells, but become enriched in intestinal or livers stem cells.

In line with the reasoning to the previous question, these variants will be shared between clone and subclone and will therefore not end up in the list of variants that have been accumulated in between the two clonal steps and that are unique to the subclone.

4) The authors should analyse and present a list of mutations that were not detectable in the parental cells but were detectable in the first clonal culture (representing de novo mutations during reprogramming and/or initial culture phase, or rare mutations inherited by the analysed clones from their parental cells).

We aim to identify those mutations that accumulate *in vitro* during the routine culturing of stem cells. To this end, we have employed an elegant experimental set-up. The mutagenic effects of the derivation process, particularly relevant for iPS cells, are beyond the scope of this study and have already been described elsewhere (Gore et al., Somatic coding mutations in human induced pluripotent stem cells, Nature 2011).

5) How many population doublings occur for the different cell types between first single cell cloning until sequencing of the clonal bulk culture?

We did not calculate the amount of doublings between the single cell step and the collection of material for WGS for the subclones, since the amount of doublings before collection of the clones is not relevant for our analyses. Any mutation that is acquired after the clonal step will have a VAF<0.3 and will therefore be excluded from the analysis.

How many population doublings occur until the second cloning step for each cell type (53-146 days)? This should be included in supplementary table 1.

We have included the requested information in supplementary table 1

6) The bulk population after the extended in vitro culture period (used for the 2nd cloning step) should also be sequenced and results should be compared to the sequencing results of the early bulk culture after 1st cloning and the bulk culture after 2nd cloning. This will allow to discriminate between de novo mutations that occurred during normal culture and mutations that occur or become selected due to the very stressful conditions during single cell cloning.

The mutations that occur during single cell cloning will be shared by all the cells in the bulk population. These will therefore also be present in the individual cells that are sequenced after the next clonal step. Sequencing of a late bulk culture therefore doesn't have an added value. We chose a lengthy in vitro culture period, to minimize the contribution of the mutations induced by the clonal step.

7) In the Material and Methods section it is mentioned that "Welch Two Sample t-tests were performed to determine whether the mutation rates differ significantly between the stem cell types". To my knowledge, more samples (in particular more than 3 iPSC samples) are required to use a T-test for analysis. In the results section, only the ANOVA test is mentioned. Which test has finally been used? The authors should comment on the choice and applicability of the applied tests.

We have used an ANOVA to test for significant differences when more than two groups were compared (Liver, intestine, and pluripotent stem cells) and we used a two-sided Student's t-test when two groups were compared (low oxygen versus high oxygen levels). We have changed the manuscript accordingly.

8) Number of three analysed iPSC subclones from only one donor may be not sufficient to be representative. At least three additional iPSC subclones from another primary clone should be included. Are the four liver and the six intestinal subclones each derived from one primary clone?

This is a valid point. We have included two additional pluripotent stem cell lines, one iPSC cell line from a different donor and the H9 ES cell line. For both pluripotent stem cell lines we generated clonal lines. Clones were cultured for 3 months to allow mutations to accumulate, followed by a 2nd clonal step to identify all the genetic variants that have been accumulated in the individual cells that gave rise to these subclones. Together with the previous results, this adds up to *in vitro* mutation accumulation data of 5 subclones from three genetically distinct pluripotent stem cell lines. The liver

and intestinal stem cell lines were derived from two different donors. In spite of their different origins (ASCs from the liver, ASCs from intestine, iPS cells from renal cells, and the ES cells of embryonic origin), all show the *in vitro* signature (e.g. many C to A changes). In our extensively revised manuscript, we have now emphasized the finding that *in vitro* culture induces specific types of mutations irrespective of cell type and we place less emphasis on the differences between the different stem cell types.

Results:

Page 3: "We found a population doubling time of ~23h for human iPS cells which is approximately twice as fast as for human liver and intestinal ASCs, that have population doubling times of ~46h and ~44h, respectively (figure 1a)". Was the level of cell death determined and taken into account for this calculation?

Population doubling time is the amount of time it takes for the cell population to reach twice its original size. Population doubling times were inferred from counts of the number of cells at different time points. It therefore takes into account both the proliferation and cell death rates, without the necessity to measure and model these separately.

Page 3: "Base pair substitutions were significantly depleted in genic regions for all three stem cell types (figure 1d)." What means „depleted“? Less than statistically expected? What is the basis for this calculation?

Indeed, with depleted we mean less than expected based on the number of mutations and the proportion of the genome that is genic. As we now explain in our materials and methods section, we have used the one-sided binomial test from the MutationalPatterns package.

Page 3: "None of the non-synonymous mutations affected known cancer genes based on the census of human cancer genes²¹ or have previously been described to confer a selective advantage over stem cells in culture^{6,14}. We are therefore confident that the observed mutations provide an unbiased insight into mutation accumulation free of selection." Lack of description as cancer gene does not necessarily mean that there is no growth advantage for a specific mutation. Also there are not many small scale mutations described so far that provide a selective advantage during culture. Therefore, it cannot be concluded that there is an "unbiased insight into mutation accumulation free of selection". In addition, it may well be that a certain mutation does not provide any selection during normal culture but during single cell cloning, for instance mutations in apoptosis-related genes.

We have removed the specific sentence from the revised version of our manuscript.

*Discussion section: The authors claim that "their approach, in contrast to previous studies, is able to discriminate mutations that originate in culture from those that have an *in vivo* origin or arise as a result of the derivation process" (see page 5, discussion).*

If the WGS data of the clonal cultures are compared with corresponding nonclonal reference sample (bulk of urothelial cells that were parental cells of the analysed iPSCs, and corresponding donor blood in case of ASCs) it is of course possible to exclude germline variants and variants that are present in the reference sample at relatively high levels (higher than detection limit of the direct WGS, which depends on the error rate of the applied Illumina sequencing approach (sequencing-by-synthesis). What is the detection level? 10%? Or even higher?

Our analysis is based on the detection of variants that are present at high frequency (VAF>0.3). For 30X coverage, this means only those variants are selected that are shared by at least nine reads, which excludes most of the false positives. Positions that were found to be variable in at least three unrelated individuals were excluded as these represent either unknown SNPs or recurring sequencing and/or calling artefacts. Targeted validations in independent experiments have demonstrated that 91% of the mutations can be validated (described in detail in Blokzijl et al 2016).

Importantly, direct WGS excludes most mutations that have been acquired over a lifetime after the first cell doublings in the early embryo and that did not become enriched in urothelial cells or blood cells during life. It has to be considered that the parental cell population, which has been used for reprogramming of iPSCs or isolation of ASCs, represents a genetic mosaic. Probably every cell has a unique signature of mutations that had been acquired over lifetime. Therefore, both, mutations that are too rare in the reference cell population but are present in the specific cell clone, and mutations that occur during the first culture period and the first cloning step, will be detectable in the bulk culture after the first cloning step. Therefore, it is not always possible to discriminate between mutations that originate in culture and those those that have an in vivo origin or arise as a result of the derivation process

See also our previous answers. Variants that have an in vivo origin or are mosaic in the bulk culture, for example because they appeared during the derivation process, will be shared between clone and subclone and will therefore not end up in the list of variants that have been accumulated in between the two clonal steps and that are unique to the subclone.

Materials and Methods section: Page 9: Sequencing and Bioinformatics: How are “callable regions” specifically defined? Which minimal coverage?

To determine the callable loci region of a sample, we used the CallableLoci tool from GATK (version 3.4.46) and considered genomic regions with a minBaseQuality score of 10, a minMappingQuality score of 10, a minDepth of 20 and a minDepthForLowMAPQ of 20. Next, we intersected the callable loci regions of the bulk, clonal and subclonal sample and only considered SBS and indel variants located in these regions. This info has now been added to the revised manuscript.

Page 9: “To identify all SNVs of the clonal cultures we used our data analysis pipeline that enables the identification of somatic SNVs with a confirmation rate of ~91% 18.” The authors should provide more information. Why 91%?

Targeted validations in independent experiments have demonstrated that 91% of the mutations can be validated (described in detail in Blokzijl et al 2016).

Page 9: "The SNV call set is further filtered on the basis of several quality parameters and dbSNP v137.b37 40." Also here more information is required!

We have now included the quality parameters in our revised manuscript: base call quality >100, minimum coverage at variant position >20 and VAF-filtering threshold of 0.3 (from Jager et al. 2018).

Page 9: "Welch Two Sample t-tests were performed to determine whether the mutation rates differ significantly between the stem cell types." I guess T-test is not appropriate if only 3 or 4 samples are available.

We have included more samples and now perform an ANOVA to determine whether the mutation rates differ significantly between the stem cell types.

Page 9: "We subsequently excluded indels that overlap with a SNV, indels that have a dbSNP ID and no COSMIC ID, and indels that are present on a blacklist of 3 unrelated samples (BED file available upon request)." What is this black list?

This approach has now been modified and the Materials and Methods section has been revised accordingly: "We excluded indels that are present in the dbSNP database to filter out known germline variants, as well as those that are present in an other sample used in this study, to exclude rare germline variants and somatic calling artifacts. From these, we only considered indels with a filter 'PASS' from VariantFiltration, with a GATK genotype quality of at least 99 and those with a REF and ALT read support between 10x and 60x in clone, subclone, and control sample (average read depth is 30x). Subsequently, we followed the same approach as with SBSs to detect the culture associated variants. Here, indels from the subclonal sample with any evidence in the respective control and clonal sample (with VAF > 0.3) were filtered out. From these, only clonal indels with a VAF support higher than 0.3 were considered."

Page 10: CNVs were called using Control-FREEC. What is the minimal coverage for those analyses? What fraction of the total genome matched this minimal coverage?

For calling structural variation including CNVs we have now used GRIDSS version 1.8.0 and post-processing from the Hartwig Medical Foundation pipeline v4.8, available at <https://github.com/hartwigmedical/pipeline>. We manually inspected all variants in the filtered somatic output using IGV to determine true-positive variant calls, and whether the structural variant accumulated *in vitro*.

Figure & Legends: Legends should always be placed below the corresponding figure, otherwise it is confusing.

We have modified the manuscript accordingly

Legend Figur 1: Should termed "Single Nucleotide Variants" instead of "mutations"; 1e) more different colours should be used for "Coding" and "non-synonymous"; it would be helpful for many readers to have a definition of (non)synonymous included in the legend f) How is

promoter-flanking defined? Everything that is not in promoters? Certain distance around promoters?

We have modified the figure and the legend accordingly. Promoter flanking regions were derived from Zerbino, D.R., Wilder, S.P., Johnson, N. et al. The Ensembl Regulatory Build. Genome Biol 16, 56 (2015) doi:10.1186/s13059-015-0621-5: " By overlapping these segmentation states, produced by unsupervised machine learning, with known genomic features, we assigned them functional labels, such as 'predicted promoter with TSS' (where TSS is transcription start site), 'predicted transcribed region', 'predicted promoter flank', 'predicted enhancer', 'CTCF enriched', 'predicted repressed', 'predicted low activity', 'predicted heterochromatin'." For further details we refer to this paper.

Conclusion: I think that the topic of acquired mutations in iPSCs and ASCs is still a very important topic, which has particular relevance for cellular therapies. Clearly, many important questions in this field remained unanswered yet.

The study of Kuijk et al. provides novel data, but suffers from methodological limitations and also partially by conclusions that cannot be drawn from the data. For publication in Nat Communications, definitely major revisions would be required.

We thank the reviewer for the comments. The reviewer's suggestions have led to major revisions that in our opinion improved our manuscript considerably.

Reviewer #3 (Remarks to the Author):

In this manuscript, Kuijk et al. analyze the burden and spectrum of mutations that arise during the in vitro culture of human pluripotent and adult stem cells. By applying an elegant study design that includes two clonal steps, the authors were able to capture and analyze the mutations that arose during a defined period of time (and a defined number of cell divisions) in subclones of three stem cell types: iPSCs, liver ASCs and intestinal ASCs. Their main findings are that the in vitro mutation rates for ASCs are ~40-fold higher than those reported in these cells in vivo; that functionally-important genomic regions are less commonly altered in all three stem cell types (as expected); and that the in vitro mutational processes are different from the in vivo mutational processes, and are characterized by C to A changes that have been previously linked to oxidative stress. This is an interesting study that takes an important step towards understanding the mutational processes that affect cultured stem cells. Such understanding is necessary to improve stem cell culture practices and ultimately enable safer use of these cells in regenerative medicine. Overall, the experiments and analyses are well-designed and carefully-performed. However, it is not clear how generalizable the findings of this study really are. A major limitation is that a single parental population was used to derive a few subclones of each stem cell type, so that eventually comparisons were performed across clonal populations that originated from a single iPSC line, a single liver ASC line and a single intestinal ASC line (i.e., n=1 per SC type). To what extent these cell lines represent their stem cell "type" is not clear, and cannot be addressed without further experiments. It would be much more informative to compare clonal cultures derived from independent parental

populations (taken from different individuals) in order to be able to assess the variability that exists within each cell type.

See also our previous answers to the comments of the other reviewers. We agree with the reviewer and have included additional pluripotent stem cell lines from two different donors and origins, iPS cells derived from renal cells and ES cells derived from preimplantation embryos. The below table gives an overview of the number of subclones that have been sequenced and from how many donors these were derived. As the reviewer acknowledges, an important message of the manuscript is that in spite of their different origins (ASCs from the liver, ASCs from intestine, iPS cells from renal cells, and the ES cells of embryonic origin), all show the *in vitro* signature (e.g. many C to A changes), indicating these findings are generalizable. In our extensively revised manuscript, we have now emphasized the finding that *in vitro* culture induces specific patterns of mutation accumulation irrespective of cell type and we place less emphasis on the differences between the different stem cell types.

Celltype	Number of donors	# subclones	Total
Liver	2	2	4
S.I.	2	3	6
PS normoxia	3	5	5
PS hypoxia	3	3	3

The manuscript could also benefit from some additional elaborations and clarifications, as described below: 1. The authors argue that their findings “demonstrate that in vivo mutational processes only play minor roles in in vitro mutation accumulation”. This is surprising and interesting, and the manuscript clearly provides evidence for in vitro mutational processes that are different from the in vivo ones. But if the role of the in vivo mutational processes is only minimal, why does hierarchical clustering based on the relative contribution of COSMIC signatures (Fig. 2c) show the in vitro ASCs to cluster together with the in vivo liver ASCs and apart from the other in vitro groups? Can this be better clarified?

We link the *in vitro* mutational signature to oxidative stress. Clustering of the *in vivo* liver samples with the *in vitro* samples may indicate that oxidative stress also plays an important role in mutation accumulation in the liver.

2. The authors state that “none of the non-synonymous mutations affected known cancer genes based on the census of human cancer genes, or have previously been described to confer a selective advantage over stem cells in culture. We are therefore confident that the observed mutations provide an unbiased insight into mutation accumulation free of selection.” I do not believe that we can be so confident that selection is not involved. First, the list of genes involved in cancer that was used for the comparison (ref 21) is a census of amplified and overexpressed human cancer genes, and is thus partial by definition (it does not cover tumor suppressors, for example). Second, the selection pressures in culture are likely to be at least somewhat different from those experienced in vivo, and somewhat tissue-type specific; however, the list of genes selected for in stem cell cultures, which was used for comparison here, is limited to pluripotent stem cells (refs 6 and 14). Can the authors compare their mutated gene list to additional/extended gene lists? This statement should also be toned down

Comparing our list with extended gene lists would indeed be possible. We have removed the conclusion from our manuscript.

3. What specific comparisons were performed in Fig. 1b,c? The legend says ANOVA, but multiple p-values appear (Fig. 1b) and the p-values are presented above specific bars (Fig. 1b,c). Are these p-values from contrast analyses? Which contrasts do the shown asterisks represent? This should be clarified in the Figures and in the Methods.

In these figures, we have performed ANOVAs. Now contrast analyses were performed. We now present one p-value.

4. Given that each stem cell type is cultured under distinct culture conditions, it is impossible to tease apart the role of the cell lineage and that of the culture conditions in shaping the mutational landscapes of the cultured cells. Can the authors comment on this point? The authors keep referring to “stem cell-specific” mutational processes but these could really be “culture conditions-specific” mutational processes. Can the authors comment on that?

The clustering of the cells by cell type could indeed have a cell intrinsic origin or the result of differences in culture methods (e.g. growth factor combinations). It is difficult to discriminate between these two possibilities and that is also beyond the scope of this study. As mentioned in our previous answers, we now place less emphasis on the differences between the stem cell types and more emphasis on the shared *in vitro* mutational signature.

5. One of the most interesting findings of the study – and certainly the finding with most practical implications – is that all cultured stem cells are enriched for C to A changes, which may be linked to oxidative stress. However, the study does not provide any direct evidence that oxidative stress indeed plays a major role in the mutational processes. An experimental demonstration that reducing ROS levels could change the rate and the spectrum of culture-acquired mutations (even if provided only for one of the SC types) would significantly increase the impact of this work.

See also our answer to the comment of reviewer one. To confirm that the C>A changes are caused by oxidative stress, we have also cultured clones of the three pluripotent stem cell lines for three months under low oxygen levels (3%), followed by a second clonal step to measure all the genetic variants in the individual cells that gave rise to the subclones. Mutation frequencies under low oxygen conditions were significantly lower than under high oxygen levels (20%). In addition, the mutation spectra of cells cultured under low oxygen conditions were significantly different from those cultured under high oxygen conditions. In particular, the relative number of C>A were greatly reduced, whereas the relative number of C>T and C>T at CpG sites were enhanced. We agree with the reviewer that these findings strengthen our conclusion that *in vitro* culture of stem cells induces mutation accumulation through oxidative stress.

Reviewers' comments:

Reviewer #1 (Remarks to the Author):

I thank the authors for thoroughly revising the manuscript. My concerns are fully addressed.

Reviewer #2 (Remarks to the Author):

In their revised manuscript "The mutational impact of culturing human pluripotent and adult stem cells" Kuijk et al. describe the comparative analysis of human induced pluripotent stem cells (iPS cells) and adult stem cells (ASCs) from intestine and liver for accumulation of de novo mutations during in vitro culture with in vivo mutations by whole genome sequencing.

The major findings include:

- 1) iPS cells, intestinal ASCs and liver ASCs have different mutation rates and accumulated 3.5 ± 0.5 , 7.2 ± 1.1 and 8.3 ± 3.6 base substitutions per population doubling, respectively.
- 2) The annual in vitro mutation accumulation rate of ASCs is ~40- fold higher than the in vivo rate of ~40 base pair substitutions per year.
- 3) Irrespective of stem cell type, mutational signatures in vitro are distinct from the in vivo mutational signature. This in vitro signature is characterized by C to A changes that have previously been linked to oxidative stress conditions.
- 4) Reducing oxygen tension in pluripotent stem cell cultures lowered the total number of mutations and in particular the number of C to A changes. This highlights the need to further optimize stem cell culture conditions in particular to reduce culture-associated oxidative stress.

Comments:

In their revised version of the manuscript, the authors properly addressed the majority of my concerns. Importantly, they removed the lack of clarity leading to a manuscript with coherent description of the methodology.

One major new aspect is the reduced mutation rate under low oxygen conditions. Although this effect had already been presumed, to my knowledge no other study provided data to prove that.

Moreover, additional lines have been analyzed, which further improved the data.

There is only one of my comments that has not been properly addressed:

Reviewer comment:

6) The bulk population after the extended in vitro culture period (used for the 2nd cloning step) should also be sequenced and results should be compared to the sequencing results of the early bulk culture after 1st cloning and the bulk culture after 2nd cloning. This will allow to discriminate between de novo mutations that occurred during normal culture and mutations that occur or become selected due to the very stressful conditions during single cell cloning.

Author's response:

The mutations that occur during single cell cloning will be shared by all the cells in the bulk population. These will therefore also be present in the individual cells that are sequenced after the next clonal step. Sequencing of a late bulk culture therefore doesn't have an added value. We chose a lengthy in vitro culture period, to minimize the contribution of the mutations induced by the clonal step.

I don't agree with this statement of the authors.

Single cell cloning with its stressful conditions is unlikely to be limited to the first cell division. The stressful conditions are caused by lack of required cell-cell contacts and lack of autocrine/paracrine

factors. It is unlikely that the required conditions that reduce cell stress (at least to a large extent) are already achieved after the very first cell division. This implicates that a single cell cloning-related increased mutation rate can be expected not only during the very first cell division after cell dissociation and single cell seeding, and that single cell cloning-related mutations will not be present in all cases in the entire cell population but may also be present in a sub-fraction of cells, only.

Subtracting the mutations present in the early bulk culture after the 1st cloning step from the mutations present in the early bulk culture after the 2nd cloning step will yield mutations that occurred during further expansion of the early bulk culture following the first cloning step (not caused by cloning-related stress) and in addition also mutations that occurred during the 2nd cloning step and the following cell expansion (potentially caused by cloning-related stress).

In contrast, subtracting the mutations present in the early bulk culture following the 1st cloning step from the mutations present in the late bulk culture prior to the 2nd cloning step will yield only mutations that occurred during further expansion of the early bulk culture after the first cloning step (not caused by cloning-related stress).

Of course, sequencing of bulk cultures always suffer from the limitation that only mutations present in a subpopulation of cells above the detection limit of the sequencing approach will be detected. On the other hand, a cloning step eliminates a majority of mutations that are present in a fraction of the bulk culture but not in the cloned cell itself.

Therefore, I still think that sequencing of the late bulk culture prior to the 2nd cloning step would provide important additional information and should enable to identify expansion-related mutations that emerged independent of the cloning-related stress, and to define their proportion among the entirety of expansion/cloning-related mutations.

Conclusion:

I think that the topic of acquired mutations in stem cells is still a very important topic, which has particular relevance for cellular therapies. Clearly, important questions in this field remained unanswered yet.

The revised manuscript of Kuijk et al. has clearly been improved and also provides important novel data. If also my last comment would be properly addressed, the manuscript should be accepted by Nat Communications.

Reviewer #3 (Remarks to the Author):

The authors addressed the Reviewers' comments satisfactorily. and the revised manuscript is considerably improved. In particular, the analysis of more pluripotent stem cell lines, and the experiments in low oxygen are important additions to the manuscript.

One issue that should be addressed prior to acceptance of the manuscript:

The revised manuscript puts much emphasis on the calculation of the likelihood that stem cell culture would result in oncogenic mutations. The authors seem to have switched the database for calling 'oncogenic mutations' (which seems now to be Tamborero et al. Genome Med 2018). However, the Methods section does not provide any information on the criteria that were used to call a mutation 'oncogenic' (there is more than one way to do that). An elaboration is required in order to correctly interpret the analysis shown in Fig. 4.

Reviewer #1 (Remarks to the Author):

Comments:

I thank the authors for thoroughly revising the manuscript. My concerns are fully addressed.

Reviewer #2 (Remarks to the Author):

Comments:

In their revised version of the manuscript, the authors properly addressed the majority of my concerns. Importantly, they removed the lack of clarity leading to a manuscript with coherent description of the methodology. One major new aspect is the reduced mutation rate under low oxygen conditions. Although this effect had already been presumed, to my knowledge no other study provided data to prove that. Moreover, additional lines have been analyzed, which further improved the data. There is only one of my comments that has not been properly addressed:

Reviewer comment:

6) The bulk population after the extended in vitro culture period (used for the 2nd cloning step) should also be sequenced and results should be compared to the sequencing results of the early bulk culture after 1st cloning and the bulk culture after 2nd cloning. This will allow to discriminate between de novo mutations that occurred during normal culture and mutations that occur or become selected due to the very stressful conditions during single cell cloning.

Author's response:

The mutations that occur during single cell cloning will be shared by all the cells in the bulk population. These will therefore also be present in the individual cells that are sequenced after the next clonal step. Sequencing of a late bulk culture therefore doesn't have an added value. We chose a lengthy in vitro culture period, to minimize the contribution of the mutations induced by the clonal step.

I don't agree with this statement of the authors.

Single cell cloning with its stressful conditions is unlikely to be limited to the first cell division. The stressful conditions are caused by lack of required cell-cell contacts and lack of autocrine/paracrine factors. It is unlikely that the required conditions that reduce cell stress (at least to a large extent) are already achieved after the very first cell division. This implicates that a single cell cloning-related increased mutation rate can be expected not only during the very first cell division after cell dissociation and single cell seeding, and that single cell cloning-related mutations will not be present in all cases in the entire cell population but may also be present in a sub-fraction of cells, only. Subtracting the mutations present in the early bulk culture after the 1st cloning step from the mutations present in the early bulk culture after the 2nd cloning step will yield mutations that occurred during further expansion of the early bulk culture following the first cloning step (not caused by cloning-related stress) and in addition also mutations that occurred during the 2nd cloning step and the following cell expansion (potentially caused by cloning-related stress). In contrast, subtracting the mutations present in the early bulk culture following the 1st cloning step from the mutations present in the late bulk culture prior to the 2nd cloning step will yield only mutations that occurred during further expansion of the early bulk culture after the first cloning step (not caused by cloning-related stress). Of course, sequencing of bulk cultures always suffer from the limitation that only mutations present in a subpopulation of cells above the detection

limit of the sequencing approach will be detected. On the other hand, a cloning step eliminates a majority of mutations that are present in a fraction of the bulk culture but not in the cloned cell itself. Therefore, I still think that sequencing of the late bulk culture prior to the 2nd cloning step would provide important additional information and should enable to identify expansion-related mutations that emerged independent of the cloning-related stress, and to define their proportion among the entirety of expansion/cloning-related mutations.

Author's response:

We acknowledge that the clonal step may cause mutations in the first few divisions, but the experiment suggested by the reviewer (compare early and late cultures of the clone and the subclone), is in our view not appropriate to detect these mutations. Mutations that arise before the first division will have a VAF of 0.5, at the 2-cell stage will have a VAF of 0.25, at the 3rd cell stage 0.125 and so on. These frequencies will not change between an early bulk culture and a late bulk culture, unless there is genetic drift or selection. Thus, a comparison between early and late cultures will only highlight genetic drift/selection, not mutations induced by the clonal step. Moreover, as input for WGS we used DNA extracted from over 100000 cells, while the sequencing was performed at on average 30X coverage, which is the equivalent of 15 diploid cells. This only enables the detection of clonal and early mutations irrespective of whether the sample is from a late or an early culture. Thus, indeed some of the detected mutations may have originated from the clonal step itself. Therefore, we chose a lengthy in vitro culture period, to minimize the contribution of the mutations induced by the clonal step. The contribution of these clonally induced mutations to total mutation load is minimal and will not have an effect on mutation patterns. This is also clear from the mutational patterns of the cells grown under 3% oxygen and the 20% oxygen. While both conditions went through a clonal and a subclonal step, the mutational patterns are clearly different from one another, indicating minimal contribution of the mutations induced by the clonal itself. While from a fundamental perspective it might be interesting to determine the mutations induced by the clonal step itself (using a different method than suggested by the reviewer), this is beyond the scope of our study because it is not very relevant for most regenerative medicine, pharmaceutical, and toxicological applications of stem cells that will mostly not involve clonal steps. Even if clonal steps are involved, the relative contribution of the clonal step to the total number of mutations at the end of the culture approaches zero and the probability that oncogenic mutations are induced by the clonal step is also close to zero. To inform the reader on the potential effect of the clonal step we now discuss this in the 2nd paragraph of the discussion section of our revised manuscript

Conclusion:

I think that the topic of acquired mutations in stem cells is still a very important topic, which has particular relevance for cellular therapies. Clearly, important questions in this field remained unanswered yet. The revised manuscript of Kuijk et al. has clearly been improved and also provides important novel data. If also my last comment would be properly addressed, the manuscript should be accepted by Nat Communications.

Reviewer #3 (Remarks to the Author):

The authors addressed the Reviewers' comments satisfactorily. and the revised manuscript is considerably improved. In particular, the analysis of more pluripotent stem cell lines, and the experiments in low oxygen are important additions to the manuscript. One issue that should be addressed prior to acceptance of the manuscript: The revised manuscript puts much

emphasis on the calculation of the likelihood that stem cell culture would result in oncogenic mutations. The authors seem to have switched the database for calling 'oncogenic mutations' (which seems now to be Tamborero et al. Genome Med 2018). However, the Methods section does not provide any information on the criteria that were used to call a mutation 'oncogenic' (there is more than one way to do that). An elaboration is required in order to correctly interpret the analysis shown in Fig. 4.

Author's response:

In the previous and the current version of our manuscript, we used a list of oncogenic mutations in driver genes from Tamborero et al. to calculate the number of mutations in the coding sequence. The list was downloaded from: <https://www.cancergenomeinterpreter.org/mutations>. The list was compiled using multiple methods as is explained in the manuscript by Tamorero et al. and also summarized on the website:

"This is a compiled inventory of mutations in cancer genes that are demonstrated to drive tumor growth or predispose to cancer. This was retrieved by combining the data contained in the DoCM (PMID:27684579), ClinVar (PMID:26582918) and OncoKB (PMID:28890946) databases as well as the results of several published experimental assays and additional manual curation efforts. We also considered as oncogenic the mutations reported to increase sensitivity to targeted drugs included in the Cancer Biomarkers Database of the Cancer Genome Interpreter. Germline variants found to predispose to cancer, which we retrieved from the ClinVar (PMID:26582918) and IARC (PMID:17311302) resources, were also included."

In the materials and methods section we now explicitly refer to Tamborero et al for details on how this list was compiled.

REVIEWERS' COMMENTS:

Reviewer #3 (Remarks to the Author):

The authors adequately addressed my concern.